# Interpreting and Boosting Dropout from a Game-Theoretic View

**Hao Zhang**
Shanghai Jiao Tong University
1603023-zh@sjtu.edu.cn

**Sen Li**
Sun Yat-sen University
lisen6@mail2.sysu.edu.cn

**Yinchao Ma**
Huazhong University of Science and Technology
u201713506@hust.edu.cn

**Mingjie Li**
Shanghai Jiao Tong University
limingjie0608@sjtu.edu.cn

**Yichen Xie**
Shanghai Jiao Tong University
xieyichen@sjtu.edu.cn

**Quanshi Zhang**[*]
Shanghai Jiao Tong University
zqs1022@sjtu.edu.cn

## Abstract

This paper aims to understand and improve the utility of the dropout operation from the perspective of game-theoretic interactions. We prove that dropout can suppress the strength of interactions between input variables of deep neural networks (DNNs). The theoretic proof is also verified by various experiments. Furthermore, we find that such interactions were strongly related to the over-fitting problem in deep learning. Thus, the utility of dropout can be regarded as decreasing interactions to alleviate the significance of over-fitting. Based on this understanding, we propose an interaction loss to further improve the utility of dropout. Experimental results have shown that the interaction loss can effectively improve the utility of dropout and boost the performance of DNNs.

## 1 Introduction

Deep neural networks (DNNs) have exhibited significant success in various tasks, but the over-fitting problem is still a considerable challenge for deep learning. Dropout is usually considered as an effective operation to alleviate the over-fitting problem of DNNs. Hinton et al. (2012); Srivastava et al. (2014) thought that dropout could encourage each unit in an intermediate-layer feature to model useful information without much dependence on other units. Konda et al. (2016) considered dropout as a specific method of data augmentation. Gal & Ghahramani (2016) proved that dropout was equivalent to the Bayesian approximation in a Gaussian process.

Our research group led by Dr. Quanshi Zhang has proposed game-theoretic interactions, including interactions of different orders (Zhang et al., 2020) and multivariate interactions (Zhang et al., 2021b). As a basic metric, the interaction can be used to explain signal-processing behaviors in trained DNNs from different perspectives. For example, we have built up a tree structure to explain hierarchical interactions between words encoded in NLP models (Zhang et al., 2021a). We also prove a close relationship between the interaction and the adversarial robustness (Ren et al., 2021) and transferability (Wang et al., 2020). Many previous methods of boosting adversarial transferability can be explained as the reduction of interactions, and the interaction can also explain the utility of the adversarial training (Ren et al., 2021).

As an extension of the system of game-theoretic interactions, in this paper, we aim to explain, model, and improve the utility of dropout from the following perspectives. First, we prove that the dropout

---

[*]Correspondence. This study is conducted under the supervision of Dr. Quanshi Zhang. zqs1022@sjtu.edu.cn. Quanshi Zhang is with the John Hopcroft Center and the MoE Key Lab of Artificial Intelligence, AI Institute, at the Shanghai Jiao Tong University, China.

operation suppresses interactions between input units encoded by DNNs. This is also verified by various experiments. To this end, the interaction is defined in game theory, as follows. Let $x$ denote the input, and let $f(x)$ denote the output of the DNN. For the $i$-th input variable, we can compute its importance value $\phi(i)$, which measures the numerical contribution of the $i$-th variable to the output $f(x)$. We notice that the importance value of the $i$-th variable would be different when we mask the $j$-th variable *w.r.t.* the case when we do not mask the $j$-th variable. Thus, the interaction between input variables $i$ and $j$ is measured as the difference $\phi_{\text{w/ }j}(i) - \phi_{\text{w/o }j}(i)$.

Second, we also discover a strong correlation between interactions of input variables and the over-fitting problem of the DNN. Specifically, the over-fitted samples usually exhibit much stronger interactions than ordinary samples.

Therefore, we consider that the utility of dropout is to alleviate the significance of over-fitting by decreasing the strength of interactions encoded by the DNN. Based on this understanding, we propose an interaction loss to further improve the utility of dropout. The interaction loss directly penalizes the interaction strength, in order to improve the performance of DNNs. The interaction loss exhibits the following two distinct advantages over the dropout operation. (1) The interaction loss explicitly controls the penalty of the interaction strength, which enables people to trade off between over-fitting and under-fitting. (2) Unlike dropout which is incompatible with the batch normalization operation (Li et al., 2019), the interaction loss can work in harmony with batch normalization. Various experimental results show that the interaction loss can boost the performance of DNNs.

Furthermore, we analyze interactions encoded by DNNs from the following three perspectives. (1) First, we discover the consistency between the sampling process in dropout (when the dropout rate $p = 0.5$) and the sampling in the computation of the Banzhaf value. The Banzhaf value (Banzhaf III, 1964) is another metric to measure the importance of each input variable in game theory. Unlike the Shapley value, the Banzhaf value is computed under the assumption that each input variable independently participates in the game with the probability 0.5. We find that the frequent inference patterns in Banzhaf interactions (Grabisch & Roubens, 1999) are also prone to be frequently sampled by dropout, thereby being stably learned. This ensures the DNN to encode smooth Banzhaf interactions. We also prove that the Banzhaf interaction is close to the aforementioned interaction, which also relates to the dropout operation with the interaction used in this paper. (2) Besides, we find that the interaction loss is better to be applied to low layers than being applied to high layers. (3) Furthermore, we decompose the overall interaction into interaction components of different orders. We visualize the strongly interacted regions within each input sample. We find out that interaction components of low orders take the main part of interactions and are suppressed by the dropout operation and the interaction loss.

Contributions of this paper can be summarized as follows. (1) We mathematically represent the dependence of feature variables using as the game-theoretic interactions, and prove that dropout can suppress the strength of interactions encoded by a DNN. In comparison, previous studies (Hinton et al., 2012; Krizhevsky et al., 2012; Srivastava et al., 2014) did not mathematically model the the dependence of feature variables or theoretically proved its relationship with the dropout. (2) We find that the over-fitted samples usually contain stronger interactions than other samples. (3) Based on this, we consider the utility of dropout is to alleviate over-fitting by decreasing the interaction. We design a novel loss function to penalize the strength of interactions, which improves the performance of DNNs. (4) We analyze the properties of interactions encoded by DNNs, and conduct comparative studies to obtain new insights into interactions encoded by DNNs.

## 2 RELATED WORK

**The dropout operation.** Dropout is an effective operation to alleviate the over-fitting problem and improve the performance of DNNs (Hinton et al., 2012). Several studies have been proposed to explain the inherent mechanism of dropout. According to (Hinton et al., 2012; Krizhevsky et al., 2012; Srivastava et al., 2014), dropout could prevent complex co-adaptation between units in intermediate layers, and could encourage each unit to encode useful representations itself. However, these studies only qualitatively analyzed the utility of dropout, instead of providing quantitative results. Wager et al. (2013) showed that dropout performed as an adaptive regularization, and established a connection to the algorithm AdaGrad. Konda et al. (2016) interpreted dropout as a kind of data augmentation in the input space, and Gal & Ghahramani (2016) proved that dropout was equiva-

lent to a Bayesian approximation in the Gaussian process. Gao et al. (2019) disentangled the dropout operation into the forward dropout and the backward dropout, and improved the performance by setting different dropout rates for the forward dropout and the backward dropout, respectively. Gomez et al. (2018) proposed the targeted dropout, which only randomly dropped variables with low activation values, and kept variables with high activation values. In comparison, we aim to explain the utility of dropout from the view of game theory. Our interaction metric quantified the interaction between all pairs of variables considering all potential contexts, which were randomly sampled from all variables. Furthermore, we propose a method to improve the utility of dropout.

**Interaction.** Previous studies have explored interactions between input variables. Bien et al. (2013) developed an algorithm to learn hierarchical pairwise interactions inside an additive model. Sorokina et al. (2008) detected the statistical interaction using an additive model-based ensemble of regression trees. Murdoch et al. (2018); Singh et al. (2018); Jin et al. (2019) proposed and extended the contextual decomposition to measure the interaction encoded by DNNs in NLP tasks. Tsang et al. (2018) measured the pairwise interaction based on the learned weights of the DNN. Tsang et al. (2020) proposed a method, namely GLIDER, to detect the feature interaction modeled by a recommender system. Janizek et al. (2020) proposed the Integrated-Hessian value to measure interactions, based on Integral Gradient (Sundararajan et al., 2017). Integral Gradient measures the importance value for each input variable *w.r.t.* the DNN. Given an input vector $x \in \mathbb{R}^n$, Integrated Hessians measures the interaction between input variables (dimensions) $x_i$ and $x_j$ as the numerical impact of $x_j$ on the importance of $x_i$. To this end, Janizek et al. (2020) used Integral Gradient to compute Integrated Hessians. Appendix J further compares Integral Hessians with the interaction proposed in this paper.

Besides interactions measured from the above views, game theory is also a typical perspective to analyze the interaction. Several studies explored the interaction based on game theory. Lundberg et al. (2018) defined the interaction between two variables based on the Shapley value for tree ensembles. Because Shapley value was considered as the unique standard method to estimate contributions of input words to the prediction score with solid theoretic foundations (Weber, 1988), this definition of interaction can be regarded to objectively reflect the collaborative/adversarial effects between variables *w.r.t* the prediction score. Furthermore, Grabisch & Roubens (1999) extended this definition to interactions among different numbers of input variables. Grabisch & Roubens (1999) also proposed the interaction based on the Banzhaf value. In comparison, the target interaction used in this paper is based on the Shapley value (Shapley, 1953). Since the Shapley is the unique metric that satisfies *the linearity property, the dummy property, the symmetry property, and the efficiency property* (Ancona et al., 2019), the interaction based on the Shapley value is usually considered as a more standard metric than the interaction based on the Banzhaf value. In this paper, we aim to explain the utility of dropout using the interaction defined in game theory. We reveal the close relationship between the strength of interactions and the over-fitting of DNNs. We also design an interaction loss to improve the performance of DNNs.

## 3 GAME-THEORETIC EXPLANATIONS OF DROPOUT

**Preliminaries: Shapley values.** The Shapley value was initially proposed by Shapley (1953) in game theory. It is considered as a unique unbiased metric that fairly allocates the numerical contribution of each player to the total reward. Given a set of players $N = \{1, 2, \cdots, n\}$, $2^N \stackrel{\text{def}}{=} \{S | S \subseteq N\}$ denotes all possible subsets of $N$. A game $f : 2^N \to \mathbb{R}$ is a function that maps from a subset to a real number. $f(S)$ is the *score* obtained by the subset $S \subseteq N$. Thus, $f(N) - f(\emptyset)$ denotes the *reward* obtained by all players in the game. The Shapley value allocates the overall reward to each player, as its numerical contribution $\phi(i|N)$, as shown in Equation (1). The Shapley value of player $i$ in the game $f$, $\phi(i|N)$, is computed as follows.

$$\sum\nolimits_{i=1}^{n} \phi(i|N) = f(N) - f(\emptyset) \,, \ \phi(i|N) = \sum\nolimits_{S \subseteq N \setminus \{i\}} P_{\text{Shapley}}\big(S | N \setminus \{i\}\big) \big[f(S \cup \{i\}) - f(S)\big] \quad (1)$$

where $P_{\text{Shapley}}(S|M) = \frac{(|M|-|S|)!|S|!}{(|M|+1)!}$ is the likelihood of $S$ being sampled, $S \subseteq M$. The Shapley value is the unique metric that satisfies *the linearity property, the dummy property, the symmetry property, and the efficiency property* (Ancona et al., 2019). We summarize these properties in Appendix A.

**Understanding DNNs via game theory.** In game theory, some players may form a coalition to compete with other players, and win a reward (Grabisch & Roubens, 1999). Accordingly, a DNN $f$ can be considered as a game, and the output of the DNN corresponds to the score $f(\cdot)$ in Equation (1).

For example, if the DNN has a scalar output, we can take this output as the score. If the DNN outputs a vector for multi-category classification, we select the classification score corresponding to the true class as the score $f(\cdot)$. Alternatively, $f$ can also be set as the loss value of the DNN.

The set of players $N$ corresponds to the set of input variables. We can analyze the interaction and the element-wise contribution at two different levels. (1) We can consider input variables (players) as the input of the entire DNN, *e.g.* pixels in images and words in sentences. In this case, the game $f$ is considered as the entire DNN. (2) Alternatively, we can also consider input variables as a set of activation units before the dropout operation. In this case, the game $f$ is considered as consequent modules of the DNN.

$S \subseteq N$ in Equation (1) denotes the context of the input variable $i$, which consists of a subset of input variables. In order to compute the network output $f(S)$, we replace variables in $N \setminus S$ with the baseline value (*e.g.* mask them), and we do not change the variables in $S$. In particular, when we consider neural activations before dropout as input variables, such activations are usually non-negative after ReLU. Thus, their baseline values are set to 0. Both (Ancona et al., 2019) and Appendix G introduce details about the baseline value. In this way, $f(\emptyset)$ measures the output score when all input variables are masked, and $f(S) - f(\emptyset)$ measures the entire reward obtained by all input variables in $S$.

**Interactions encoded by DNNs.** In this section, we introduce how to use the interaction defined in game theory to explain DNNs. Two input variables may interact with each other to contribute to the output of a DNN. Let us suppose input variables $i$ and $j$ have an interaction. In other words, the contribution of $i$ and $j$ when they work jointly is different with the case when they work individually. For example, in the sentence *he is a green hand*, the word *green* and the word *hand* have a strong interaction, because the words *green* and *hand* contribute to the person's identity jointly, rather than independently. In this case, we can consider these two input variables to form a certain inference pattern as a singleton player $S_{ij} = \{i, j\}$. Thus, this DNN can be considered to have only $(n-1)$ input variables, $N' = N \setminus \{i, j\} \cup S_{ij}$, *i.e.* $S_{ij}$ is supposed to be always absent or present simultaneously as a constituent. In this way, the interaction $I(i, j)$ between input variables $i$ and $j$ is defined by Grabisch & Roubens (1999), as the contribution increase of $S_{ij}$ when input variables $i$ and $j$ cooperate with each other *w.r.t.* the case when $i$ and $j$ work individually, as follows.

$$I(i,j) \stackrel{\text{def}}{=} \phi(S_{ij}|N') - \big[\phi(i|N \setminus \{j\}) + \phi(j|N \setminus \{i\})\big] = \sum\nolimits_{S \subseteq N \setminus \{i,j\}} P_{\text{Shapley}}\big(S|N \setminus \{i,j\}\big) \Delta f(S, i, j) \quad (2)$$

where $\Delta f(S, i, j) \stackrel{\text{def}}{=} f(S \cup \{i, j\}) - f(S \cup \{j\}) - f(S \cup \{i\}) + f(S)$. $\phi(i|N \setminus \{j\})$ and $\phi(j|N \setminus \{i\})$ correspond to the contribution to the DNN output when $i$ and $j$ work individually. Theoretically, $I(i, j)$ is also equal to the change of the variable $i$'s Shapley value when we mask another input variable $j$ *w.r.t.* the case when we do not mask $j$.

If $I(i, j) > 0$, input variables $i$ and $j$ cooperate with each other for a higher output value. Whereas, if $I(i, j) < 0$, $i$ and $j$ have a negative/adversarial effect. The strength of the interaction can be computed as the absolute value of the interaction, *i.e.* $|I(i, j)|$. We find that the overall interaction $I(i, j)$ can be decomposed into interaction components with different orders $s$. We use the multi-order interaction defined in (Zhang et al., 2020), as follows.

$$I(i,j) = \sum_{s=0}^{n-2} \Big[\frac{I^{(s)}(i,j)}{n-1}\Big], \qquad I^{(s)}(i,j) \stackrel{\text{def}}{=} \mathbb{E}_{S \subseteq N \setminus \{i,j\}, |S|=s}\Big[\Delta f(S, i, j)\Big] \quad (3)$$

where $s$ denote the size of the context $S$ for the interaction. We use $I^{(s)}(i, j)$ to represent the $s$-order interaction between input variables $i$ and $j$. $I^{(s)}(i, j)$ reflects the average interaction between input variables $i$ and $j$ among all contexts $S$ with $s$ input variables. For example, when $s$ is small, $I^{(s)}(i, j)$ measures the interaction relying on inference patterns consisting of very few input variables, *i.e.* the interaction depends on a small context. When $s$ is large, $I^{(s)}(i, j)$ corresponds to the interaction relying on inference patterns consisting of a large number of input variables, *i.e.* the interaction depends on the context of a large scale.

*Visualization of interactions in each sample, which are encoded by the DNN:* There exists a specific interaction between each pair of pixels, which boosts the difficulty of visualization. In order to simplify the visualization, we divide the original image into $16 \times 16$ grids, and we only visualize the strength of interactions between each grid $g$ and its neighboring grids $g'$ as $\text{Color}(g) = \mathbb{E}_{g' \in \text{neighbor}(g)}[|I(g, g')|]$. Figure 1 visualizes the interaction strength within the image in the CelebA

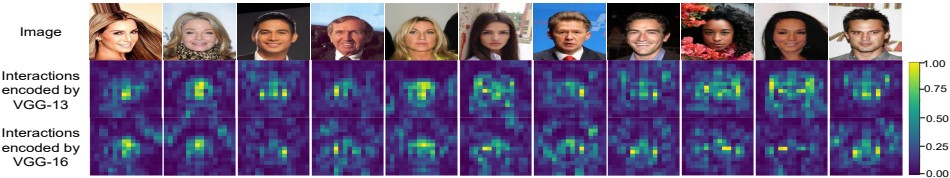

Figure 1: Visualization of the interaction strength encoded by DNNs.

dataset (Liu et al., 2015), which are normalized to the range $[0, 1]$. Grids on the face usually contain more significant interactions with neighboring grids than grids in the background.

**Proof of the relationship between dropout and the interaction.** In this section, we aim to mathematically prove that dropout is an effective method to suppress the interaction strength encoded by DNNs. Given the context $S$, let us consider its subset $T \subseteq S$, which forms a coalition to represent a specific inference pattern $T \cup \{i, j\}$. Note that for dropout, the context refers to activation units in the intermediate-layer feature without semantic meanings. Nevertheless, we just consider $i, j$ as pixels as a toy example to illustrate the basic idea, in order to simplify the introduction. For example, let $S$ represent the face, and let $T \cup \{i, j\}$ represent pixels of an eye in the face. Let $R^T(i, j)$ quantify the marginal reward obtained from the inference pattern of an eye. All interaction effects from smaller coalitions $T' \subsetneq T$ are removed from $R^T(i, j)$.

According to the above example, Let $T' \subsetneq T$ correspond to the pupil inside the eye. Then, $R^{T'}(i, j)$ measures the marginal reward benefited from the existence of the pupil $T' \cup \{i, j\}$, while $R^T(i, j)$ represents the marginal benefit from the existence of the entire eye, in which the reward from the pupil has been removed. *I.e.* the inference pattern $T \cup \{i, j\}$ can be exclusively triggered by the co-occurrence of all pixels in the eye, but cannot be triggered by a subset of pixels in the pupil $T' \cup \{i, j\}$. The benefit from the pupil pattern has been removed from $R^T(i, j)$. Thus, the $s$-order interaction can be decomposed into components *w.r.t.* all inference patterns $T \cup \{i, j\}, T \subseteq S$.

$$I^{(s)}(i, j) = \mathbb{E}_{S \subseteq N \setminus \{i,j\}, |S|=s} \left[ \sum_{T \subseteq S} R^T(i, j) \right] = \sum_{0 \le q \le s} \binom{s}{q} J^{(q)}(i, j) = \sum_{0 \le q \le s} \Gamma^{(q)}(i, j|s) \quad (4)$$

where $J^q(i, j) \stackrel{\text{def}}{=} \mathbb{E}_{T \subseteq N \setminus \{i,j\}, |T|=q}[R^T(i, j)]$ denotes the average interaction between $i$ and $j$ given all potential inference patterns $T \cup \{i, j\}$ with a fixed inference pattern size $|T| = q$; $\Gamma^{(q)}(i, j|s) \stackrel{\text{def}}{=} \binom{s}{q} J^{(q)}(i, j)$. The computation of $R^T(i, j)$ and the proof of Equation (4) are provided in Appendices D and B, respectively.

However, when input variables in $N$ are randomly removed by the dropout operation, the computation of $I^{(s)}_{\text{dropout}}(i, j)$ only involves a subset of inference patterns consisting of variables that are not dropped. Let the dropout rate be $(1 - p), p \in [0, 1]$, and let $S' \subseteq S$ denote the input variables that remain in the context $S$ after the dropout operation. Let us consider cases when $r$ activation units remain after the dropout operation, *i.e.* $|S'| = r$. Then, the average interaction $I^{(s)}_{\text{dropout}}(i, j)$ in these cases can be computed as follows.

$$I^{(s)}_{\text{dropout}}(i, j) = \mathbb{E}_{S \subseteq N \setminus \{i,j\}, |S|=s} \left[ \mathbb{E}_{S' \subseteq S, |S'|=r} \left( \sum_{T \subseteq S'} R^T(i, j) \right) \right] = \sum_{0 \le q \le r} \Gamma^{(q)}(i, j|r) \quad (5)$$

The interaction only comprises the marginal reward from the inference patterns consisting of at most $r \sim \text{B}(s, p)$ variables, where $\text{B}(s, p)$ is the binomial distribution with the sample number $s$ and the sample rate $p$. Since $r = |S'| \le s$, we have

$$1 \ge \frac{\Gamma^{(1)}(i, j|r)}{\Gamma^{(1)}(i, j|s)} \ge \cdots \ge \frac{\Gamma^{(r)}(i, j|r)}{\Gamma^{(r)}(i, j|s)} \ge 0, \qquad \frac{I^{(s)}_{\text{dropout}}(i, j)}{I^{(s)}(i, j)} = \frac{\sum_{0 \le q \le r} \Gamma^{(q)}(i, j|r)}{\sum_{0 \le q \le s} \Gamma^{(q)}(i, j|s)} \le 1. \quad (6)$$

We assume that most $\Gamma^{(q)}(i, j|s), 0 \le q \le s$ share the same signal. In this way, we can get Equation (6, right) based on law of large numbers, which shows that the number of inference patterns usually significantly decreases when we use dropout to remove $s - r, r \sim \text{B}(s, p)$, activation units. Please see Appendix C for the proof.

*Experimental verification 1: Inference patterns with more activation units are less likely to be sampled when we use dropout, thereby being more vulnerable to the dropout operation, according to Equation (6, left).* Inspired by this, we conducted experiments to explore the following two

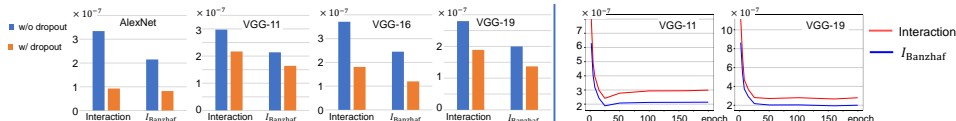

Figure 2: Curves of strength of interaction components with different orders during the learning process. All interactions were computed between activation units before dropout. $\Delta I^{(s)} = |I^{(s)}_{\text{w/ dropout}}| - |I^{(s)}_{\text{w/o dropout}}|$ denotes the change of interaction strength.

Figure 3: (left) The decrease of interactions caused by the dropout operation. (right) Close relationship between the target interaction $\mathbb{E}_{\text{image}}[\mathbb{E}_{(i,j)\in\text{image}}|I(i,j)|]$ and the interaction based on the Banzhaf value $\mathbb{E}_{\text{image}}[\mathbb{E}_{(i,j)\in\text{image}}|I_{\text{Banzhaf}}(i,j)|]$. We show the change of the two types of interactions during the learning process. The target interaction is highly related to the interaction based on Banzhaf value. Here, all interactions were computed between activation units before dropout.

terms, (1) which component $\{I^{(s)}(i,j)\}(s = 0, ..., n-2)$ took the main part of the overall interaction strength among all interaction components with different orders; (2) which interaction component was mainly penalized by dropout.

To this end, the strength of the $s$-order interaction component was averaged over images, *i.e.* $I^{(s)} = \mathbb{E}_{\text{image}}[\mathbb{E}_{(i,j)\in\text{image}}(|I^{(s)}(i,j)|)]$. Specifically, when we analyzed the strength of interaction components, we selected the following orders, $s = 0.1n, 0.3n, ..., 0.9n$. Note that we randomly sampled the value of $s$ from $[0.0, 0.2n]$ for each context $S$ to approximate the interaction component with the order $s = 0.1n$. We also used the similar approximation for $s = 0.3n, ..., 0.9n$. Figure 2 shows curves of different interaction components[1] within VGG-11/19 (Simonyan & Zisserman, 2015) learned on the CIFAR-10 (Krizhevsky & Hinton, 2009) dataset with and without the dropout operation. We found that the interaction components with low orders took the main part of the interaction.

*Experimental verification 2:* We also conducted experiments to illustrate how dropout suppressed the interaction modeled by DNNs, which was a verification of Equation (6, right). In experiments, we trained AlexNet (Krizhevsky et al., 2012), and VGG-11/16/19 on CIFAR-10 with and without the dropout. Figure 3(left) compares the strength of interactions[1] encoded by DNNs, which were learned with or without the dropout operation. When we learned DNNs with dropout, we set the dropout rate as $0.5$. Please see Appendix H.1 for experiments on different dropout rates. We averaged the strength of interactions over images, *i.e.* $I = \mathbb{E}_{\text{image}}[\mathbb{E}_{(i,j)\in\text{image}}(|I(i,j)|)]$, where $I(i,j)$ was obtained according to Equation (2).

Note that accurately computing the interaction of two input variables was an NP-hard problem. Thus, we applied a sampling-based method (Castro et al., 2009) to approximate the strength of interactions. Furthermore, we conducted an experiment to explore the accuracy of the interactions approximated via the sampling-based method. Please see Appendix K for details. Castro et al. (2009) proposed a method to approximate the Shapley value, which can be extended to the approximation of the interaction. We found that dropout could effectively suppress the strength of the interaction, which verified the above proof.

**Property: The sampling process in dropout is the same as the computation in the Banzhaf value.** In this section, we aim to show that the sampling process in dropout (when the dropout rate is 0.5) is similar to the sampling in the computation of the Banzhaf value. Just like the Shapley value, the Banzhaf value (Banzhaf III, 1964) is another typical metric to measure importance of each input variable in game theory. Unlike the Shapley value, the Banzhaf value is computed under the assumption that each input variable independently participates in the game with the probability $0.5$. The Banzhaf value is computed as $\psi(i|N) = \sum_{S\subseteq N\setminus\{i\}} P_{\text{Banzhaf}}(S|N\setminus\{i\})[f(S\cup\{i\}) - f(S)]$, where $P_{\text{Banzhaf}}(S|N\setminus\{i\}) = 0.5^{n-1}$ is the likelihood of $S$ being sampled. The form of the Banzhaf value is similar to that of the Shapley value in Equation (1), but the sampling weight of the Banzhaf value

---

[1]For fair comparison, we normalize the value of interaction using the range of output scores of DNNs. Please see Appendix E for more details.

| Dataset | Model | Ordinary | Over-fitted |
|---------|-------|----------|-------------|
| MNIST | RN-44 | $2.17\times10^{-3}$ | $\mathbf{3.64\times10^{-3}}$ |
| Tiny-ImageNet | RN-34 | $2.57\times10^{-3}$ | $\mathbf{2.89\times10^{-3}}$ |
| CelebA | RN-34 | $6.46\times10^{-3}$ | $\mathbf{1.17\times10^{-2}}$ |

Table 1: Comparison of the interaction strength between the over-fitted samples and ordinary samples. DNNs usually encode more interactions for the over-fitted samples than ordinary samples.

| DNN | Dataset | Loss$_{\text{interaction}}$ | | Dropout | |
|-----|---------|------------|-------------|-----------|------------|
| | | low layers | high layers | low layers | high layers |
| AlexNet | CIFAR-10 | **69.6** | 67.2 | 67.5 | 68.4 |
| VGG-13 | CIFAR-10 | **66.2** | 61.4 | 60.9 | 59.7 |
| VGG-16 | CIFAR-10 | **64.9** | 62.6 | 63.0 | 59.5 |
| VGG-19 | Tiny-ImageNet | **45.2** | 37.4 | 32.6 | 37.0 |
| VGG-16 | CelebA | **94.6** | 93.9 | 92.4 | 94.0 |

Table 2: Classification accuracy of DNNs when we applied the dropout operation and the interaction loss at different positions.

$P_{\text{Banzhaf}}(S|N\setminus\{i\})$ is different from $P_{\text{Shapley}}(S|N\setminus\{i\})$ of the Shapley value. For dropout with the dropout rate 0.5, let $n$ be the number of input variables, and let $S$ be the units not dropped in $N\setminus\{i\}$. Then, the likelihood of $S$ not being dropped is given as $P_{\text{dropout}}(S|N\setminus\{i\}) = 0.5^{|S|}0.5^{n-|S|-1} = P_{\text{Banzhaf}}(S|N\setminus\{i\})$. In this way, dropout usually generates activation units $S$ following $P_{\text{Banzhaf}}(S|N\setminus\{i\})$. Therefore, the frequent inference patterns in the computation of the Banzhaf value are also frequently generated by dropout, thereby being reliably learned by the DNN. This makes that $f(S\cup\{i\}) - f(S)$ in the computation of the Banzhaf value can be sophisticatedly modeled without lots of outlier values. Thus, the dropout can be considered as a smooth factor in terms of the Banzhaf value.

Grabisch & Roubens (1999) defined the interaction based on the Banzhaf value as $I_{\text{Banzhaf}}(i,j) = \sum_{S\subseteq N\setminus\{i,j\}} P_{\text{Banzhaf}}(S|N\setminus\{i,j\})\Delta f(S,i,j)$, just like Equation (2). Thus, dropout can also be considered as a smooth factor in the computation of the Banzhaf interaction. Figure 3(right) shows that the target interaction used in this study is closely related to the Banzhaf interaction, which potentially connects the target interaction based on the Shapley value to the dropout operation.

## 4 UTILITY OF DROPOUT & IMPROVEMENT OF THE DROPOUT UTILITY

**Close relationship between the strength of interactions and over-fitting.** In this section, we conducted various experiments to explore the relationship between the interaction strength and the over-fitted samples. We noticed that in the classification task, the over-fitted samples were usually outliers. To this end, we assigned 5% training samples with random incorrect labels in MNIST (Lecun et al., 1998), CelebA(Liu et al., 2015), and Tiny ImageNet (Le & Yang, 2015). When the DNN was well-trained on such training data with an almost zero training loss, then we took samples with incorrect labels as the over-fitted samples. We trained ResNet-34 (RN-34) (He et al., 2016) for the classification task using the Tiny ImageNet dataset and the CelebA dataset, and trained ResNet-44 (RN-44) for the classification task using the MNIST dataset. For each trained DNN, we divided input images into grids, and computed the average interaction strength between grids on the over-fitted samples and ordinary samples, respectively. As Table 1 shows, the over-fitted samples usually contained stronger interactions[1] than ordinary samples.

**Understanding of dropout.** We have proved that dropout can decrease the strength of interactions encoded in DNNs in Section 3. Besides, previous paragraphs have shown that the over-fitted samples usually encode more interactions by the DNN than other samples. Therefore, we consider the utility of the dropout operation is to decrease interaction strength to reduce the significance of over-fitting.

**Further improvement of the utility of dropout using the interaction loss.** Based on the above understanding, we develop the interaction loss as an improvement of the utility of dropout. To this end, we apply this loss to learn DNNs, in order to boost the performance, as follows.

$$\text{Loss} = \text{Loss}_{\text{classification}} + \lambda\text{Loss}_{\text{interaction}}, \tag{7}$$

where $\lambda > 0$ is the weight of the interaction loss. Let us focus on an intermediate-layer feature after the ReLU operation $\mathbf{h} \in \mathbb{R}^n$ as $n$ activation units. We aim to suppress interactions between any two units $i, j \in N$. Thus, the interaction loss can be formulated as $\text{Loss}_{\text{interaction}} = \mathbb{E}_{i,j\in N, i\neq j}[|I(i,j)|]$. The baseline value in the computation of $I(i,j)$ is set to 0, as is explained in the paragraph *understanding DNNs via game theory*, Section 3.

$$\text{Loss}_{\text{interaction}} = \mathbb{E}_{i,j\in N, i\neq j}\left[|I(i,j)|\right] = \mathbb{E}_{i,j\in N, i\neq j}\left[\left|\sum_{S\subseteq N\setminus\{i,j\}} P_{\text{Shapley}}(S|N\setminus\{i,j\})\left[\Delta f(S,i,j)\right]\right|\right] \tag{8}$$

However, it is extremely computational expensive to use the above interaction loss to train the DNN. Therefore, we propose the following approximation of the interaction loss, which is implemented in a batch manner, instead of averaging all pairs of activation units. Specifically, we sample disjoint

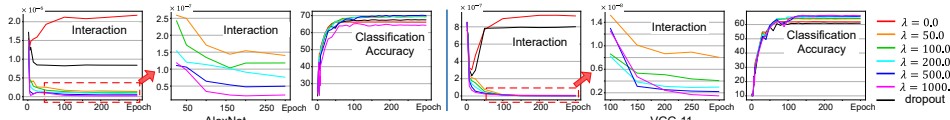

Figure 4: Curves of interaction strength encoded by DNNs trained using different weights of the interaction loss. The interaction strength decreases along with the increase of the weight $\lambda$. For fair comparison, we computed the interaction between activation units before the layer, in which we used the interaction loss or the dropout operation.

subsets of units $A, B \subsetneq N, A \cap B = \emptyset$, instead of sampling two single units $i, j$, to compute for the interaction loss. Here, we can regard $A$ and $B$ as batches of the sampled units $\{i\}, \{j\}$, respectively. Accordingly, the context $S$ is disjoint with $A$ and $B$, *i.e.* $S \subseteq N \setminus A \setminus B$. We can get the following approximation of $\text{Loss}_{\text{interaction}}$. Please see Appendix F for the proof.

$$\text{Loss}'_{\text{interaction}} = \mathbb{E}_{A,B \subsetneq N, A \cap B = \emptyset, |A|=|B|=\alpha n} \left\{ \mathbb{E}_r \left[ \mathbb{E}_{|S|=r, S \subseteq N \setminus A \setminus B} \left[ \Delta f(S, A, B)^2 \right] \right] \right\} \tag{9}$$

where $\Delta f(S, A, B) \stackrel{\text{def}}{=} f(S \cup A \cup B) - f(S \cup A) - f(S \cup B) + f(S)$, and $\alpha$ is a small positive constant. In this paper, we set $\alpha = 0.05$. Figure 4(left) verifies the proposed approximated interaction loss can successfully reduce the interaction defined in Equation (2).

**Advantages.** Compared with dropout, the interaction loss exhibits following advantages.

*Advantage 1:* The interaction loss enables people to explicitly control the penalty of the interaction strength by adjusting the weight $\lambda$ in Equation (7). The explicit control of the interaction strength is crucial, because it is important to make a careful balance between over-fitting and under-fitting during the learning of the DNN. The weight $\lambda$ needs to be carefully set, since it is usually difficult to trade off between over-fitting and under-fitting in deep learning. A large value of $\lambda$ may lead to under-fitting, while a small value of $\lambda$ may increase the risk of over-fitting. In comparison, the dropout rate can control the strength of the dropout, when we consider the utility of the dropout as a penalty of interactions. However, we find that the interaction loss exhibits a stronger power to reduce the interaction than the dropout. To this end, we have conducted experiments to compare the strength of the dropout with the strength of the interaction loss, which verified the conclusion on the strength of the dropout. Please see Appendix H.1 for details.

*Advantage 2:* Due to the disharmony between dropout and the batch normalization (Li et al., 2019), people usually cannot apply dropout in the DNN with batch normalization. In comparison, the interaction loss is compatible with batch normalization. For example, we compute $\text{Loss}_{\text{interaction}}$ using a new track, which is different from the track of computing $\text{Loss}_{\text{classification}}$. In other words, we do not update the parameter in batch normalization layers when we compute $\text{Loss}_{\text{interaction}}$. Please see Appendix G for more discussions.

**Experiments:** We trained DNNs for the classification based on the CIFAR-10 dataset (Krizhevsky & Hinton, 2009), the classification based on the Tiny ImageNet dataset (Le & Yang, 2015), and the face attribution estimation based on the CelebA dataset (Liu et al., 2015). We trained six types of DNNs, including AlexNet[2] (Krizhevsky et al., 2012), VGG-11/13/16 (Simonyan & Zisserman, 2015), and ResNet-18/34 (RN-18/34) (He et al., 2016). Note that we only randomly sampled 10% training data in the CIFAR-10 dataset and the Tiny ImageNet dataset, and randomly sampled training data form the CelebA dataset. Besides, we sampled over 80% units to generate the context $S$ in experiments using ResNets, in order to ensure the stability of training. This was because a DNN usually showed a low significance of over-fitting, when it was learned on the complete dataset. We only sampled a small proportion of training images to better show the over-fitting problem. In this experiment, when we trained DNNs with dropout, we set the dropout rate as 0.5. Please see Appendix H.1 for experiments on different dropout rates. Furthermore, we trained the BERT (Devlin et al., 2018) using the SST-2 dataset (Socher et al., 2013) in order to verify the effectiveness of the interaction loss on linguistic data. Please see Appendix H.2 for more details.

● *Learning DNNs with the interaction loss.* We trained the DNNs with different weights of the interaction loss, and evaluated their testing accuracy. Table 3 shows that the accuracy of DNNs trained with different values of $\lambda$ and the accuracy of DNNs trained with dropout. The accuracy of DNNs

---

[2]We slightly adjusted the structure of DNNs to adapt them to input images with sizes $32 \times 32$ and $64 \times 64$, according to the classic and standard implementation of (pytorch-cifar100, 2020).

**CIFAR-10 dataset**

| $\lambda$ | AlexNet[2] | VGG-11[2] | VGG-13[2] | VGG-16[2] |
|---|---|---|---|---|
| 0.0 | 66.2 | 61.9 | 60.8 | 62.0 |
| 50.0 | 69.2 | 63.9 | 64.0 | 63.8 |
| 100.0 | 69.6 | 64.3 | 65.4 | 64.5 |
| 200.0 | 69.6 | 65.3 | 65.9 | 64.7 |
| 500.0 | **70.0** | 65.9 | **66.2** | **64.9** |
| 1000.0 | 64.3 | **66.3** | 66.0 | 64.5 |
| Dropout | 67.5 | 60.9 | 60.9 | 63.0 |

**Tiny ImageNet**

| $\lambda$ | RN-18[2] | RN-34[2] | $\lambda$ | VGG-16 | VGG-19 |
|---|---|---|---|---|---|
| 0.0 | 48.8 | 45.6 | 0.0 | 33.4 | 37.6 |
| 0.001 | 50.0 | 48.4 | 50.0 | 38.4 | 38.2 |
| 0.003 | 49.6 | 49.0 | 100.0 | 38.0 | 38.6 |
| 0.01 | **52.2** | **49.6** | 200.0 | 38.2 | 39.0 |
| 0.03 | 50.4 | 48.8 | 500.0 | **42.8** | 41.8 |
| | | | 1000.0 | 40.8 | **45.2** |
| Dropout | 47.4 | 46.0 | Dropout | 36.8 | 32.6 |

**Gender estimation**

| $\lambda$ | VGG-13 | VGG-16 | $\lambda$ | RN-18 |
|---|---|---|---|---|
| 0.0 | 94.6 | 93.7 | 0.0 | 92.7 |
| 5.0 | 94.8 | 93.8 | 0.001 | 93.0 |
| 10.0 | 94.7 | **94.6** | 0.003 | **93.1** |
| 20.0 | **94.9** | 94.1 | 0.01 | 93.0 |
| 50.0 | 94.7 | 94.08 | 0.03 | 92.9 |
| 100.0 | 94.7 | 94.3 | | |
| Dropout | 94.6 | 92.4 | Dropout | 92.1 |

Over-fitting ↑ / Under-fitting ↓

Table 3: Classification accuracy when the DNNs are controlled from over-fitting to under-fitting.

usually increased along with the $\lambda$ at first, and then decreased when the value of $\lambda$ continued to increase. This could be explained as the interaction loss alleviated the significance of over-fitting when the weight $\lambda$ was small, and then caused under-fitting when the weight $\lambda$ was large. Furthermore, the DNN trained using the interaction loss usually outperformed the DNN trained using the dropout, when the weight $\lambda$ was properly selected. This verified that the interaction loss would improve the utility of the dropout.

Figure 4 shows curves of the interaction and the accuracy of DNNs during the learning process with different weights $\lambda$ of the interaction loss. First, we found that the interaction decreased along with the increase of the weight $\lambda$, which verified the effectiveness of the interaction loss. Second, when the weight $\lambda$ was properly selected, the DNN trained with the interaction loss usually outperformed the DNN trained with dropout. The interaction of the DNN trained using the interaction loss was much lower than the interaction of the DNN trained using the dropout. In other words, the interaction loss was more effective to suppress the interaction and boost the performance than the dropout. In particular, Table 3 shows that the performance of ResNets sometimes dropped when these DNNs were learned with dropout. This phenomenon also verified the conclusion in (Li et al., 2019), *i.e.* dropout was not compatible with batch normalization. In comparison, learning with the interaction loss did not suffer from the use of batch normalization.

• *Exploring the suitable position for the interaction loss and the dropout operation.* We learned the AlexNet and VGG-13/16 using the CIFAR-10 dataset, learned the VGG-19 using the Tiny ImageNet dataset, and learned the VGG-16 using the CelebA dataset. For each DNN, we put the dropout operation and the interaction loss in the low convolutional layer (before the 3rd/5th convolutional layer of the AlexNet/VGGs) and the high fully-connected layer (before the 2nd fully-connected layer), respectively, and compared their performance. As Table 2 shows, the interaction loss was better to be applied to the feature of a low convolutional layer. In comparison, we did not find out a clear principle about the suitable position for dropout.

**Discussions about advantages of the interaction loss.** As is mentioned above, the interaction loss exhibits two advantages over dropout. Table 3 shows that the interaction loss enabled people to explicitly control the interaction strength. When the $\lambda$ value increased, DNNs were controlled from over-fitting to under-fitting. Both the over-fitting problem and the under-fitting problem would hurt the performance. However, this explicit control could not be achieved by dropout. Furthermore, unlike dropout, the interaction loss was compatible with batch normalization. As Table 3 shows, the use of dropout decreased the classification accuracy of ResNet-18 (RN-18) on the Tiny ImageNet dataset by 1.4%, and decreased the classification accuracy of ResNet-18 (RN-18) on the CelebA dataset by 0.6%. In comparison, the use of the interaction loss does not suffer from the batch normalization operation. For fair comparison, the interaction loss and dropout were put in the same position of the DNN.

## 5 CONCLUSION

In this paper, we explain, model, and improve the utility of the dropout operation using game theory. We prove that dropout can reduce the strength of interactions, thereby improving the performance of DNNs. Experimental results have verified this conclusion. Furthermore, based on the close relationship between the interaction and over-fitting, we propose an interaction loss to directly penalize the strength of interactions, in order to improve the utility of dropout. We find that the interaction loss can reduce the interaction strength effectively and further boost the performance of DNNs.

ACKNOWLEDGMENTS

This work is partially supported by National Natural Science Foundation of China (61906120 and U19B2043).

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

## A    FOUR DESIRABLE PROPERTIES OF THE SHAPLEY VALUE

In this section, we mainly introduce the four desirable properties of the Shapley value mentioned in Section 3, including *the linearity property, the dummy property, the symmetry property and the efficiency property* (Ancona et al., 2019).

**Linearity property:** Given three games $f$, $g$, and $h$. If the reward of the game $f$ satisfies $f(S) = g(S) + h(S)$, then the Shapley value of each player $i \in N$ in the game $f$ is the sum of Shapley values of the player $i$ in the game $g$ and $h$, *i.e.* $\phi_f(i|N) = \phi_g(i|N) + \phi_h(i|N)$.

**Dummy property:** In a game $f$, If a player $i$ satisfies $\forall S \subseteq N \setminus \{i\}, f(S \cup \{i\}) = f(S) + f(\{i\})$, then this player is defined as the dummy player. The dummy player $i$ satisfies $\phi(i|N) = f(\{i\}) - f(\emptyset)$, *i.e.* the dummy player has no interaction with other players in $N$.

**Symmetry property:** Given two player $i$ and $j$ in a game $f$, if $\forall S \subseteq N \setminus \{i,j\}, f(S \cup \{i\}) = f(S \cup \{j\})$, then $\phi(i|N) = \phi(j|N)$.

**Efficiency property:** For a game $f$, the overall reward can be distributed to each player in the game. *I.e.* $\sum_{i \in N} \phi(i|N) = f(N) - f(\emptyset)$, where $f(\emptyset)$ is the obtained reward when no player participates in the game.

## B    PROOF OF EQUATION (4)

This section provides the detailed proof for Equation (4) in Section 3. The $s$-order interaction can be decomposed as follows.

$$I^{(s)}(i,j) = \mathop{\mathbb{E}}_{\substack{S \subseteq N \setminus \{i,j\} \\ |S|=s}} \left[ \sum_{T \subseteq S} R^T(i,j) \right] \tag{10}$$

$$= \frac{1}{\binom{n-2}{s}} \sum_{\substack{S \subseteq N \setminus \{i,j\} \\ |S|=s}} \left( \sum_{T \subseteq S} R^T(i,j) \right) \tag{11}$$

$$= \frac{1}{\binom{n-2}{s}} \sum_{\substack{S \subseteq N \setminus \{i,j\} \\ |S|=s}} \left[ \sum_{0 \le q \le s} \sum_{\substack{T \subseteq S \\ |T|=q}} R^T(i,j) \right] \tag{12}$$

$$= \frac{1}{\binom{n-2}{s}} \sum_{0 \le q \le s} \left[ \sum_{\substack{S \subseteq N \setminus \{i,j\} \\ |S|=s}} \sum_{\substack{T \subseteq S \\ |T|=q}} R^T(i,j) \right] \tag{13}$$

$$= \frac{1}{\binom{n-2}{s}} \sum_{0 \le q \le s} \left[ \sum_{\substack{T \subseteq N \setminus \{i,j\} \\ |T|=q}} \sum_{\substack{T' \subseteq N \setminus \{i,j\} \setminus T \\ |T'|=s-q}} R^T(i,j) \right] \quad \begin{matrix} \% \text{ let } S = T \cup T' \\ \text{and } T \cap T' = \emptyset. \end{matrix} \tag{14}$$

$$= \frac{1}{\binom{n-2}{s}} \sum_{0 \le q \le s} \left\{ \sum_{\substack{T \subseteq N \setminus \{i,j\} \\ |T|=q}} \left[ \binom{n-2-q}{s-q} R^T(i,j) \right] \right\} \tag{15}$$

$$= \frac{1}{\binom{n-2}{s}} \sum_{0 \le q \le s} \left[ \binom{n-q-2}{s-q} \binom{n-2}{q} \underbrace{\mathop{\mathbb{E}}_{\substack{T \subseteq N \setminus \{i,j\} \\ |T|=q}} R^T(i,j)}_{J^q(i,j)} \right] \tag{16}$$

$$= \sum_{0 \le q \le s} \binom{s}{q} J^q(i,j) = \sum_{0 \le q \le s} \Gamma^{(q)}(i,j|s) \tag{17}$$

In Equation (17), $J^q(i,j) = \mathbb{E}_{T \subseteq N \setminus \{i,j\}, |T|=q}[R^T(i,j)]$ denotes the average interaction between $i$ and $j$ given all potential inference patterns $T \cup \{i,j\}$ with a fixed inference pattern size $|T| = q$.

## C  PROOF OF EQUATION (5) AND EQUATION (6, LEFT)

This section gives the detailed proof for Equation (5) and Equation (6, left) in Section 3. We first prove Equation (5). The interaction of $i$ and $j$ with dropout, $I_{\text{dropout}}^{(s)}(i,j)$, can be computed as

$$I_{\text{dropout}}^{(s)}(i,j) = \mathop{\mathbb{E}}_{\substack{S \subseteq N \setminus \{i,j\} \\ |S|=s}} \left[ \mathop{\mathbb{E}}_{\substack{S' \subseteq S \\ |S'|=r}} \left( \sum_{T \subseteq S'} R^T(i,j) \right) \right] \tag{18}$$

$$= \mathop{\mathbb{E}}_{\substack{S \subseteq N \setminus \{i,j\} \\ |S|=s}} \left[ \mathop{\mathbb{E}}_{\substack{S' \subseteq S \\ |S'|=r}} \left( \sum_{T \subseteq S'} R^T(i,j) \right) \right] \tag{19}$$

$$= \frac{1}{\binom{n-2}{s}} \sum_{\substack{S \subseteq N \setminus \{i,j\} \\ |S|=s}} \frac{1}{\binom{s}{r}} \sum_{\substack{S' \subseteq S \\ |S'|=r}} \left( \sum_{T \subseteq S'} R^T(i,j) \right) \tag{20}$$

$$= \frac{1}{\binom{n-2}{s}} \frac{1}{\binom{s}{r}} \sum_{\substack{S' \subseteq N \setminus \{i,j\} \\ |S'|=r}} \sum_{\substack{S'' \subseteq N \setminus \{i,j\} \setminus S' \\ |S''|=s-r}} \left( \sum_{T \subseteq S'} R^T(i,j) \right) \quad \begin{array}{l} \text{\% similar to Equation} \\ \text{(13)} \sim \text{Equation (16).} \end{array} \tag{21}$$

$$= \frac{1}{\binom{n-2}{s}} \frac{1}{\binom{s}{r}} \binom{n-2-r}{s-r} \sum_{\substack{S' \subseteq N \setminus \{i,j\} \\ |S'|=r}} \left( \sum_{T \subseteq S'} R^T(i,j) \right) \tag{22}$$

$$= \frac{1}{\binom{n-2}{r}} \sum_{\substack{S' \subseteq N \setminus \{i,j\} \\ |S'|=r}} \left( \sum_{T \subseteq S'} R^T(i,j) \right) \tag{23}$$

$$= \mathop{\mathbb{E}}_{\substack{S' \subseteq N \setminus \{i,j\} \\ |S'|=r}} \left( \sum_{T \subseteq S'} R^T(i,j) \right) \tag{24}$$

$$= \sum_{0 \leq q \leq r} \binom{r}{q} J^q(i,j) \qquad \text{\% similar to the proof of Equation (14).} \tag{25}$$

Thus, Equation (5) holds. In order to prove Equation (6, left), we aim to prove $\forall r \leq s$ and $1 \leq q \leq r-1$, we have $1 \geq \frac{\Gamma^{(q)}(i,j|r)}{\Gamma^{(q)}(i,j|s)} \geq \frac{\Gamma^{(q+1)}(i,j|r)}{\Gamma^{(q+1)}(i,j|s)} \geq 0$. Next, we prove the three inequalities step by step.

First, $\frac{\Gamma^{(q)}(i,j|r)}{\Gamma^{(q)}(i,j|s)} = \frac{\binom{r}{q} J^{(q)}(i,j)}{\binom{s}{q} J^{(q)}(i,j)} = \frac{r!(s-q)!q!}{(r-q)!q!s!} = \frac{r!(s-q)!}{(r-q)!s!} = \frac{r \cdot (r-1) \cdots (r-q+1)}{s \cdot (s-1) \cdots (s-q+1)} \leq 1$.

Next, $\frac{\Gamma^{(q)}(i,j|r)}{\Gamma^{(q)}(i,j|s)} \Big/ \frac{\Gamma^{(q+1)}(i,j|r)}{\Gamma^{(q+1)}(i,j|s)} = \frac{r \cdot (r-1) \cdots (r-q+1)}{s \cdot (s-1) \cdots (s-q+1)} \Big/ \frac{r \cdot (r-1) \cdots (r-(q+1)+1)}{s \cdot (s-1) \cdots (s-(q+1)+1)} = \frac{s-q}{r-q} \geq 1$.

Then, $\frac{\Gamma^{(q+1)}(i,j|r)}{\Gamma^{(q+1)}(i,j|s)} = \frac{r-q}{s-q} \geq 0$.

Thus, we prove Equation (6, left).

## D  THE COMPUTATION OF $R^T(i,j)$

This section gives a recursive formulation of $R^T(i,j)$ mentioned in Section 3. $R^T(i,j)$ measures the marginal reward of the inference pattern $T \cup \{i,j\}$, where all interaction effects from smaller inference patterns $T' \subsetneqq T$ are removed from $R^T(i,j)$. In this way, $R^T(i,j)$ can be computed as follows.

$$R^T(i,j) \stackrel{\text{def}}{=} \Delta f(T,i,j) - \sum_{T' \subsetneqq T} R^{T'}(i,j) \tag{26}$$

Next, we prove the correctness of the formulation of $R^T(i,j)$. In other words, we aim to prove $I^{(s)}(i,j) = \mathbb{E}_{S \subseteq N \setminus \{i,j\}, |S|=s} \left[ \sum_{T \subseteq S} R^T(i,j) \right]$ in Equation (4).

$$\text{RHS} = \mathbb{E}_{S \subseteq N \setminus \{i,j\}, |S|=s} \Big[ \underbrace{R^S(i,j)}_{\substack{\text{the marginal reward obtained from} \\ \text{the inference pattern } S \cup \{i,j\}}} + \underbrace{\sum_{T \subsetneq S} R^T(i,j)}_{\substack{\text{the marginal reward obtained from the} \\ \text{inference patterns smaller than } S \cup \{i,j\}}} \Big] \qquad (27)$$

$$= \mathbb{E}_{S \subseteq N \setminus \{i,j\}, |S|=s} \left[ \Delta f(S,i,j) \right] \qquad \% \text{ according to Equation (26).} \qquad (28)$$

$$= I^{(s)}(i,j) = \text{LHS} \qquad \% \text{ according to Equation (3)} \qquad (29)$$

Thus, we prove the correctness of the recursive formulation of $R^T(i,j)$.

## E  THE COMPUTATION OF THE STRENGTH OF INTERACTIONS

In implementation, we randomly sampled 10 images for the computation. We divided each image into $16 \times 16$ grids in experiments to further boost the computation efficiency. We consider each grid as an input variable, and computed the interaction between two grids. We randomly sampled 80 pairs of grids for computation of the interaction. We averaged the strength of interaction over all sampled pairs of grids as the result of the interaction. Besides, for fair comparison of interactions among different DNNs, we normalized the value of interaction using the range of output score of DNNs. *I.e.* $I_{\text{norm}} = I/Y$, where $Y$ was the normalization term that reflected the range of the output score of the DNN. If the DNN was learned for multi-category classification, then $Y = |\mathbb{E}_{x \in X}[f(x)_{y^*} - \mathbb{E}_{y \neq y^*}[f(x)_y]]|$, where $x$ was sampled from the training set $X$, $y^*$ was the ground-truth label of $x$, and $f(x)_y$ denoted the value of the $y$-th dimension in the output score $f(x)$. If the DNN was learned for binary classification and the output score was a scalar, then $Y = |\mathbb{E}_{x \in X_{\text{positive}}}[f(x)] - \mathbb{E}_{x \in X_{\text{negative}}}[f(x)]|$, where $X_{\text{positive}}$ and $X_{\text{negative}}$ represent the set of positive samples and the set of negative samples, respectively.

## F  PROOF OF THE APPROXIMATION OF THE INTERACTION LOSS

This section gives the proof for the trustworthiness of the approximated interaction loss $\text{Loss}'_{\text{interaction}}$ in Equation (9). The relationship between Equation (8) and Equation (9) is proved as follows.

According to Equation (3), Equation (8), and Equation (**??**), the interaction loss can be approximated by sampling-based method, *i.e.*

$$\text{Loss}_{\text{interaction}} = \mathbb{E}_{i,j \in N}[|I(i,j)|] = \mathbb{E}_{i,j \in N}[|I'(i,j)|] = \mathbb{E}_{i,j \in N}\left[ \left| \mathbb{E}_s \left[ \mathbb{E}_{|S|=s, S \subseteq N}[\Delta f(S,i,j)] \right] \right| \right] \tag{30}$$

Let $\mu$ denote the strength of the interaction $I(i,j)$, *i.e.* $|I(i,j)| = \mu \geq 0$. We assume that $\Delta f(S,i,j)$ follows the Gaussian distribution, *i.e.* $\Delta f(S,i,j) \sim \text{Gaussian}(\hat{\mu}, \sigma^2)$. $\hat{\mu} = I(i,j)$. In this way, given a specific pair of units $i$ and $j$, we have

$$\begin{aligned} P(\Delta f(S,i,j), \mu) &= C \cdot \exp(-\frac{(\Delta f(S,i,j) - \mu)^2}{2\sigma^2})p(\mu) \\ P(\Delta f(S,i,j), -\mu) &= C \cdot \exp(-\frac{(\Delta f(S,i,j) + \mu)^2}{2\sigma^2})p(-\mu), \end{aligned} \tag{31}$$

where $C = \frac{1}{\sqrt{2\pi}\sigma}$ is a constant value. Thus, given the value of $\Delta f(S, i, j)$, the likelihood of $I(i, j) = +\mu$ is given as

$$
\begin{aligned}
P(\mu | \Delta f(S, i, j)) &= \frac{P(\Delta f(S, i, j), \mu)}{P(\Delta f(S, i, j), \mu) + P(\Delta f(S, i, j), -\mu)} \\
&= \frac{C \cdot \exp(-\frac{(\Delta f(S,i,j)-\mu)^2}{2\sigma^2})p(\mu)}{C \cdot \exp(-\frac{(\Delta f(S,i,j)-\mu)^2}{2\sigma^2})p(\mu) + C \cdot \exp(-\frac{(\Delta f(S,i,j)+\mu)^2}{2\sigma^2})p(-\mu)} \quad \% \ p(\mu) = p(-\mu) \\
&= \frac{\exp[-\frac{1}{2\sigma^2}(\Delta f(S, i, j) - \mu)^2]}{\exp[-\frac{1}{2\sigma^2}(\Delta f(S, i, j) - \mu)^2] + \exp[-\frac{1}{2\sigma^2}(\Delta f(S, i, j) + \mu)^2]} \\
&= \frac{1}{1 + \exp(-\frac{2\Delta f(S,i,j)\mu}{\sigma^2})} \\
&= \text{sigmoid}\left(\frac{2\Delta f(S, i, j)\mu}{\sigma^2}\right)
\end{aligned}
$$
(32)

To simplify the story, let us first consider some easy cases. During the learning process, if $P(\mu | \Delta f(S, i, j)) \approx 1$, then it indicates $I(i, j) = \mu > 0$. In this case, we need to decrease $\Delta f(S, i, j)$ to decrease $|I(i, j)|$. To this end, we need to set the gradient of $\Delta f(S, i, j)$ to be 1. In comparison, if $P(\mu | \Delta f(S, i, j)) \approx 0$, then it indicates $I(i, j) = -\mu < 0$, and we need to increase $\Delta f(S, i, j)$ to reduce $|I(i, j)|$. To this end, we need to set the gradient of $\Delta f(S, i, j)$ to be $-1$. In this way, for other cases, we can consider the probability $P(\mu | \Delta f(S, i, j))$ as the probability of setting the gradient of $\Delta f(S, i, j)$ to 1, and consider $1 - P(\mu | \Delta f(S, i, j))$ as the probability of setting the gradient of $\Delta f(S, i, j)$ to -1. Therefore, the expectation of the gradient can be computed as $P(\mu | \Delta f(S, i, j)) \cdot 1 + (1 - P(\mu | \Delta f(S, i, j))) \cdot (-1) = 2P(\mu | \Delta f(S, i, j)) - 1$. Furthermore, the gradient of $\Delta f(S, i, j)$ can be approximated as

$$
\begin{aligned}
2P(\mu | \Delta f(S, i, j)) - 1 &= 2 \cdot \text{sigmoid}\left(\frac{2\Delta f(S, i, j)\mu}{\sigma^2}\right) - 1 \\
&= \frac{2}{1 + \exp(-\frac{2\Delta f(S,i,j)\mu}{\sigma^2})} - 1 \quad \% \text{ We assume that } \Delta f(S, i, j) \text{ is close to zero} \\
&\approx 2\left(\frac{1}{2} + \frac{\mu}{2\sigma^2}\Delta f(S, i, j)\right) - 1 \\
&= \frac{\mu}{\sigma^2}\Delta f(S, i, j).
\end{aligned}
$$
(33)

Thus, the interaction loss can be rewritten as follows.

$$
\begin{aligned}
\text{Loss}_{\text{interaction}} &\approx \mathbb{E}_{i,j \in N}\left[\mathbb{E}_s\left[\mathbb{E}_{|S|=s, S \subseteq N}\left[\frac{\mu}{2\sigma^2}\Delta f(S, i, j)^2\right]\right]\right] \\
&\approx \mathbb{E}_{A,B \subsetneq N, A \cap B = \emptyset, |A|=|B|=\alpha n}\left\{\mathbb{E}_{i \in A, j \in B}\left[\mathbb{E}_s\left[\mathbb{E}_{|S|=s, S \subseteq N}\left[\frac{\mu}{2\sigma^2}\Delta f(S, i, j)^2\right]\right]\right]\right\} \\
&= \mathbb{E}_{A,B \subsetneq N, A \cap B = \emptyset, |A|=|B|=\alpha n}\left\{\mathbb{E}_s\left[\mathbb{E}_{|S|=s, S \subseteq N}\left[\frac{\mu}{2\alpha^2 n^2 \sigma^2}\Delta f(S, A, B)^2\right]\right]\right\}
\end{aligned}
$$
(34)

where $\frac{\mu}{2\alpha^2 n^2 \sigma^2}$ is a positive constant. Therefore, we can use Equation (9) to approximate the interaction loss.

## G  DETAILS OF THE INTERACTION AND THE INTERACTION LOSS

**Implementation of the interaction loss:** In this section, we aim to introduce the implementation details of the interaction loss. Since the computation of the interaction loss in Equation (8) is expensive, we propose Equation (9) as an approximation of the interaction loss.

As the Equation (9) shows, we need to sample subsets of units $A$, $B$, and $S$ for the computation of the interaction loss. For $A$ and $B$, we randomly select 5% units from the intermediate-layer feature, respectively, and $A \cap B = \emptyset$. As for the subset $S$, we sampled units from $N \setminus A \setminus B$.

Specifically, we first uniformly select a sampling rate from the range $[0, 1]$, and sample the subset $S$ with this sampling rate. When we apply the interaction loss to DNNs with batch normalization layers (such as ResNets), we need to compute the interaction loss using a different track with the computation of the classification loss. *I.e.* when we compute the classification loss, parameters in the batch normalization layers are normally updated. In comparison, when we compute the interaction loss, we directly use the parameters in batch normalization layers without updating their values. In this way, we can apply the interaction loss to DNNs without causing disharmony with batch normalization layers.

**The selection of the output score:** In this paper, we consider a DNN $f$ as a game, and take the output score of the DNN as the reward score. Since the reward score is usually a scalar value, for the DNN with a vectorized output, we usually take the output score before the softmax layer corresponding to the true class as the reward score. Specifically, for DNNs with batch normalization layers, such as ResNets, we take the output score after the softmax layer corresponding to the true class as the reward score.

**Details of the datasets and training settings in Section 4:** In the first experiment of Section 4, we used datasets with incorrect labels to reveal the relationship between the interaction and the over-fitting of DNNs. For the Tiny ImageNet dataset, we used all images from the first ten categories for the classification task, and each category contained 500 images. For the MNIST dataset, we randomly selected 10% images for training the DNN. In this case, each category contains 600 images. For the CelebA dataset, we randomly selected 1% training samples for the estimation of the gender. In order to build datasets with the over-fitted samples, we randomly selected 5% samples of the training samples, and replaced their labels with a randomly selected incorrect labels. In this case, when the DNN was well-learned and had a training loss close to zero, samples with incorrect labels could be considered as the over-fitted samples. In comparison, samples with correct labels were ordinary samples. Besides, in this experiment, the strength of interaction was normalized by the normalization term of each individual image.

In the following experiment of Section 4, we explored the utility of the interaction loss. We trained DNNs using the CIFAR-10 dataset, the Tiny ImageNet dataset and the CelebA dataset. In order to make the DNN suffer from the over-fitting problem, for the CIFAR-10 dataset and the Tiny ImageNet dataset, we randomly generated 10% training samples to train the DNN. More specifically, we only used the first 10 categories for the classification task. Besides, for the CelebA dataset, we used 1% training samples to train the DNN for gender estimation. In these experiments, the strength of interactions was normalized by the normalization term over all training samples.

For fair comparison, the interaction loss and dropout were put in the same position of the DNN. Specifically, for AlexNet, we put the interaction loss or dropout before the third convolutional layer. For VGGs, we put the interaction loss or dropout behind the second block, *i.e.* behind the second ReLU layer of VGG-11 and behind the forth ReLU layer of VGG-13/16/19. We put the interaction loss or dropout behind the first block for ResNet-18 (RN-18), and behind the second block for ResNet-34 (RN-34). When we put the interaction loss or dropout at a high layer, we always put the interaction loss or dropout behind the first fully-connected layer.

**The baseline value:** In this section, we introduce details about the baseline value, which was proposed by (Ancona et al., 2019). Ancona et al. (2019) proposed a polynomial-time algorithm for Shapley values approximation to explain deep neural networks. When we estimate the attribution of input variables and the interaction between input variables, we usually use a baseline value to simulate the absence of variables. Many methods of estimating attributions require the definition of a baseline value as the anchor value corresponding to the absence of the variable. Setting an input variable to the baseline value indicates this input variable is toggled off. Let us focus on the task of object classification. We follow the settings in (Ancona et al., 2019). If we aim to measure the interactions between input pixels, the baseline value is defined as the average pixel value among all samples. When we measure the interaction between activation units before the dropout operation, the baseline value is defined as zero.

| | | Dropout rates | | | | | | Weights of Loss$_{interaction}$ | | | | |
|---|---|---|---|---|---|---|---|---|---|---|---|---|
| | | 0.0 | 0.1 | 0.3 | 0.5 | 0.7 | 0.9 | 50.0 | 100.0 | 200.0 | 500.0 | 1000.0 |
| Accuracy | AlexNet | 66.2 | 67.1 | 66.6 | 67.5 | 66.1 | 57.0 | 69.2 | 69.6 | 69.6 | **70.0** | 64.3 |
| | VGG-11 | 61.9 | 61.7 | 61.5 | 60.9 | 58.1 | 28.5 | 63.9 | 64.3 | 65.3 | 65.9 | **66.3** |
| Interaction($\times 10^{-7}$) | AlexNet | 21.73 | 15.90 | 8.77 | 8.39 | 5.24 | 3.93 | 1.40 | 1.18 | 0.76 | 0.50 | **0.23** |
| | VGG-11 | 9.26 | 9.21 | 9.15 | 8.01 | 6.17 | 2.69 | 0.08 | 0.04 | 0.03 | 0.02 | **0.01** |

Table 4: (left) Classification accuracy and the interaction of DNNs with dropouts when we tried different dropout rates; (right) classification accuracy and the interaction of DNNs trained with different weights $\lambda$ of the interaction loss.

| | Dropout | Weights of Loss$_{interaction}$ | | | | |
|---|---|---|---|---|---|---|
| | | 0.0 | 0.5 | 1.0 | 5.0 | 10.0 |
| Accuracy | 77.1 | 76.3 | 77.1 | **78.0** | 77.0 | 76.5 |
| Interaction ($\times 10^{-5}$) | 24.06 | 36.89 | 0.17 | 0.15 | 0.09 | **0.08** |

Table 5: Classification accuracy and the interaction of the BERT with the dropout (dropout rate=0.5), and Classification accuracy and the interaction of BERTs with different weights $\lambda$ of the interaction loss.

| | 3rd conv | 1st fc | 2nd fc | 3rd fc |
|---|---|---|---|---|
| AlexNet | 67.5 | 69.4 | 68.4 | 67.6 |
| VGG-11 | 60.9 | 58.1 | 57.6 | 60.9 |

Table 6: Classification accuracy of DNNs when we applied the dropout operation before different layers.

| | Dropout | Loss$_{interaction}$ |
|---|---|---|
| AlexNet | 11min | 23min |
| VGG-11 | 12min | 29min |

Table 7: The time cost of training DNNs using Loss$_{interaction}$ and the time cost of training DNNs with the dropout.

# H   MORE RESULTS

## H.1   THE COMPARISON OF DROPOUT USING DIFFERENT DROPOUT RATES.

In this section, we conducted new experiments to further study the performance of different dropout rates. We trained AlexNet and VGG-11 using the CIFAR-10 dataset. We set the dropout rate as $0.1, 0.3, 0.5, 0.7, 0.9$, respectively, and applied the dropout operation before the 3rd convolutional layers of the AlexNet/VGG-11, which was the same as in experiments corresponding to Table 3. Table 4 shows the classification accuracy of different DNNs, as well as the strength of interactions modeled by DNNs. We found that the dropout rate of 0.9 usually penalized more interactions than the dropout operations with other dropout rates. Nevertheless, DNNs trained using the interaction loss still decrease more interactions than DNNs trained with the dropout in most cases. In addition, the DNN trained with a proper setting of $\lambda$ outperformed all DNNs with the dropout.

## H.2   EXPERIMENTS ON LINGUISTIC DATA

In this section, we applied our interaction loss to another type of data, *i.e.* the linguistic data. We trained the BERT model (Devlin et al., 2018) on the SST-2 (Socher et al., 2013) dataset for the binary sentiment classification task. Just like experiments in the CIFAR-10 dataset and the Tiny ImageNet dataset, we randomly sampled $10\%$ training samples. This was because the DNN showed a low over-fitting level, when it was trained using the complete dataset. We set the dropout rate as 0.5, and applied the dropout/interaction loss after embedding layers of the BERT. We designed two baselines. The first baseline was the DNN trained with the traditional dropout. The second baseline was the DNN trained without the dropout or the interaction loss. Table 5 shows that the interaction loss successfully decreased the interaction modeled by the BERT, and outperformed two baselines. Therefore, the interaction loss was a more effective method to control the interaction, and boost the performance of DNNs than the dropout.

## H.3   THE COMPARISON BETWEEN DIFFERENT POSITIONS TO APPLY THE DROPOUT

In order to verify that the dropout is applicable to convolutional layers, in this section, we conducted an experiment to apply the dropout on various fully-connected layers and the convolutional layer.

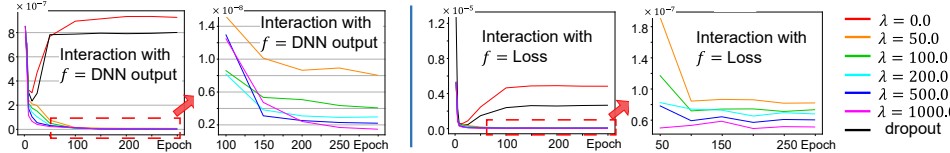

Figure 5: Curves of interactions when we set $f =$ DNN output and when we set $f = Loss$, respectively. The interaction with $f = Loss$ had similar behaviors with the interaction with $f =$ DNN output.

Specifically, we applied the dropout before the 1st/2nd/3rd fully-connected layers and before the 3rd convolutional layer of the AlexNet and VGG-11. In this experiment, we set the dropout rate as 0.5. Table 6 shows the accuracy of DNNs with different positions, to which the dropout was applied. We found that DNNs trained with dropout on the convolutional layer had similar performance with DNNs trained with dropout on fully-connected layers. Thus, dropout was applicable to convolutional layers.

### H.4 THE COMPARISON OF COMPUTATIONAL TIME BETWEEN DNNS TRAINED USING DROPOUT AND LOSS_INTERACTION

In this section, we compared the time of learning DNNs with dropout and the time of learning DNNs using the interaction loss. We trained AlexNet and VGG-11 using the CIFAR-10 dataset on a GPU of GeForce GTX-1080Ti. The experimental setting was the same as that in experiments corresponding to Table 3 and Figure 4. As Table 7 shows, our interactions loss increased the time cost, but it was still of specific value considering its good performance and the harmony with the batch-normalization operation.

### H.5 THE COMPARISON BETWEEN TAKING THE OUTPUT AS THE SCORE $f$, AND TAKING THE LOSS VALUE AS THE SCORE $f$

In this section, we aim to measure the interaction when we used the loss value of the DNN as the score $f$. We trained the VGG-11 using the CIFAR-10 dataset. Then, we compared the interaction when we set $f = Loss$ with the interaction when we set $f$ as the output score of the DNN. As Figure 5 shows, the interaction with $f = Loss$ had similar behaviors with the interaction with the network output as $f$. Therefore, the choice of the score $f$ did not affect the conclusion that the interaction loss could effectively decrease interactions encoded in the DNN.

## I DETAILS FOR VISUALIZING INTERACTIONS

This section provides the implementation details to visualize interactions modeled by DNNs. We used the CelebA dataset for visualization. There were $224 \times 224$ pixels in images of CelebA, which made it expensive to compute and visualize the interaction between any pairs of pixels. Thus, we divided the image into $16 \times 16$ grids, each grid contained $14 \times 14$ pixels. We considered these grids as input variables, and computed the interaction between any two adjacent grids. We sampled 100 times to approximate the interaction between two grids. For a specific grid, we averaged the interactions between the grid with its adjacent grids, and used the strength of the averaged interaction as the attribution value of this grid.

## J COMPARISON BETWEEN OUR INTERACTION AND THE INTEGRAL HESSIANS

The Hessian matrix is another typical way to define interactions. However, the standard way of using the Hessian matrix to define interactions is not to directly use the raw Hessian value $\nabla_{i,j}^2 L$, but to use the Integrated-Hessian value that has been proposed by (Janizek et al., 2020).

| | Interaction $\times 10^{-4}$ |
|---|---|
| Raw Hessian | 5.09 |
| Integrated Hessian | 0.81 |
| Our interaction | 5.76 |

Table 8: Comparison of different types of interactions, including the raw Hessian value, the Integrated-Hessian value, and the interaction value defined in this paper.

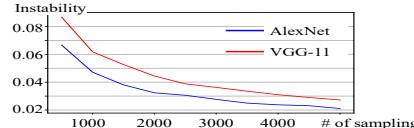

Figure 6: Instability of the interaction approximated with different sampling numbers. The instability of the approximated interaction decreased along with the increase of the sampling number.

Janizek et al. (2020) proposed the Integrated-Hessian value to measure interactions, based on Integral Gradient (Sundararajan et al., 2017). Integral Gradient is used to compute an importance value for each input variable w.r.t. the DNN. Given an input vector $x \in \mathbb{R}^n$, Integrated Hessians measures the interaction between input variables (dimensions) $x_i$ and $x_j$ as the numerical impact of $x_j$ on the importance of $x_i$. To this end, Janizek et al. (2020) used Integral Gradient to compute Integrated Hessians. More specifically, given a DNN $f$ and two input variables $x_i$ and $x_j$, the interaction between $x_i$ and $x_j$ is measured as follows.

$$I(i,j) = (x_i - x_i')(x_j - x_j') \times \int_{\beta=0}^{1} \int_{\alpha=0}^{1} \alpha\beta \frac{\partial^2 f(x' + \alpha\beta(x - x'))}{\partial x_i \partial x_j} d\alpha d\beta,$$

where $x'$ is a baseline image.

Theoretically, the Integrated-Hessian value is not equal to our interaction metric, and has different theoretic foundations with our interaction metric. Our interaction metric defines the interaction based on the Shapley value (a game-theoretic importance), In comparison, the importance value used to define the Integrated-Hessian value is computed using the Integral Gradient. Strictly speaking, the Integral-Gradient importance is highly related but not equal to the Shapley value (importance). The Shapley value is the unique importance that satisfies *the linearity property, the dummy property, the symmetry property, and the efficiency property* (Shapley, 1953; Weber, 1988; Ancona et al., 2019).

Compared to the raw Hessian value $\nabla_{i,j}^2 L$, the Integrated-Hessian value integrates over multiple high-order gradients to generate the final Integrated-Hessian value. Unlike our interaction metric, the computation of Integrated-Hessian values requires the DNN to use the SoftPlus operation to replace the ReLU operation. The computation of our interaction metric does not have such a requirement.

To this end, we conducted an experiment to show the difference between the raw Hessian value $\nabla_{i,j}^2 L$, the Integrated-Hessian value (Janizek et al., 2020), and our interaction metric. We implemented the Integrated Hessian using the code released by (suinleelab, 2020). We directly used the code and the DNN architecture provided by (suinleelab, 2020) to train the DNN on the MNIST dataset, where all ReLU layers in the DNN were replaced with the SoftPlus operation (this was required by (Janizek et al., 2020) for the computation of the Integrated-Hessian value). Given the trained DNN, we computed the strength of interactions based on the raw Hessian value, the Integrated-Hessian value, and the interaction value defined in this paper, respectively. As Table 8 shows, these metrics generated fully different interaction values, because these interaction metrics defined different types of interactions from fully different perspectives.

## K  EVALUATION OF THE ACCURACY OF THE APPROXIMATED INTERACTION

In this section, we conducted an experiment to explore the accuracy of the interactions approximated via the sampling-based method (Castro et al., 2009). More specifically, given a certain image and certain sampling number, we measured the instability of the average interaction strength $I_{\text{image}} = \mathbb{E}_{(i,j) \in \text{image}}(|I(i,j)|)$, when we repeatedly computed the interaction for multiple times considering the randomness in the sampling process. This instability value indicated whether we obtained similar interactions with different sampling states. The instability *w.r.t.* the interaction $I_{\text{image}}$ is computed as $\frac{\mathbb{E}_{u,v:u \neq v}|I_{\text{image}}^{(u)} - I_{\text{image}}^{(v)}|}{\mathbb{E}_w |I_{\text{image}}^{(w)}|}$, where $I_{\text{image}}^{(u)}$ denotes the interaction result computed in the $u$-th time. Then, we computed the average instability value over interactions of 100 pairs of input variables in 10 images to measure the accuracy of the approximated interaction. A high insta-

bility indicated a low accuracy. We used the trained AlexNet and VGG-11 to compute the instability of interactions. We computed the instability with different sampling numbers. Figure 6 shows that the instability of the approximated interaction decreased along with the increase of the sampling number. We found that when the sampling number was large than $500$, the instability of the approximated interaction was less than $0.1$, which indicated that the approximated interaction was stable and accurate enough. Thus, we set the sampling number as $500$ in all other experiments in this paper. Meanwhile, computing the interaction using such a sampling number was far more efficient than the original NP-hard computation of the interaction.

