# OpenReview forum: "Interpreting and Boosting Dropout from a Game-Theoretic View"
_ICLR.cc/2021/Conference — ICLR 2021 Poster_

### Official Review · AnonReviewer4 · 2020-10-28
**A new perspective of game theory to understand dropout**

**Rating:** 5
**Confidence:** 1

**Review:**

Summary：
The paper proves that dropout can suppress the strength of interactions between input variables from the perspective of game theory. It further improves the utility of dropout by introducing an explicit interaction loss. Experimental results verify the theoretic proof and the effectiveness of the proposed loss.

Strengths:
1. The paper introduces a new perspective of game theory to understand dropout.
2. Experiments are conducted on various datasets to support the theoretic proof and the proposed interaction loss

Concerns:
1. Although I have no background in game theory, I try my best to understand the terminology and the analysis. However, I do not have the ability to verify the correctness of its proof. Thus, I cannot evaluate the main contribution of this paper. For experimental results, the conclusion that dropout suppressing the input interactions is not a new story.
2. It would be more interesting if the author can further explain the disharmony between dropout and bn from the perspective of game theory.

---

> ### Author Response · Authors · 2020-11-20
> **Response to Reviewer #4**
>
> Thank you very much for your careful review and constructive comments. We try our best to answer all your concerns.
>
> ---
>
> Q1: “For experiment results, the conclusion that dropout suppressing the input interactions is not a new story.”
>
> A: A good question, but compared to previous work, our research has made substantial breakthroughs in the theoretical proof of the conclusion. Previous studies (Hinton et al., 2012; Krizhevsky et al., 2012; Srivastava et al., 2014)  claimed that the dropout would decrease the dependence between feature variables just based on their intuitive understanding of the dropout. However, these studies did not mathematically model the variable dependence or theoretically proved its relationship with the dropout.
>
> In comparison, we model the dependence using the interaction based on game theory, and build up a mathematical connection between the dependence and the dropout. Please see the last paragraph of the introduction section for details.
>
> Besides, we propose the interaction loss, which overcomes shortcomings of the dropout. Nevertheless, we have revised the paper to further clarify this issue. Please see the first two paragraphs on Page 8 for details.
>
> ---
>
> Q2: Suggest to explain the disharmony between dropout and bn from the perspective of game theory.
>
> A: Thanks. The disharmony between dropout and batch normalization is caused by the fact that the dropout operation affects the statistical variance learned by batch normalization, which leads to the inconsistency of variances in dropout and batch normalization (Li et al., 2019). Nevertheless, in this paper, we aim to use game theory to explain the dropout, instead of explaining the disharmony between the dropout and batch normalization.

---

### Official Review · AnonReviewer1 · 2020-10-29
**New way of measuring dropout's effect on interaction strength, but weak experimental validation and lacking in clarity**

**Rating:** 7
**Confidence:** 5

**Review:**

Summary:
This paper analyzes the effect of dropout on interaction between units in a neural network. The strength of the interaction is measured using a metric that is used in game theory to quantify interaction between players in a co-operative game. The paper shows that dropout reduces high-order interaction (as measured by this metric), and that reduction in interaction is correlated with better generalization. The paper introduces a new regularizer that explicitly minimizes the metric and claims that using this regularizer instead of dropout has some advantages.

Pros:
- The idea that dropout reduces overfitting by breaking up complex co-adaptations and regularizing interactions is widely believed to be true. However, this paper tries to explicitly quantify the amount of interaction and presents theoretical and experimental evidence that interaction reduces as a result of having dropout.

Cons:
- The proposed metric is hard to compute exactly since it requires summing over exponentially many terms, each term requiring a forward prop through the network.
- The assumptions made in computing this metric approximately seem unclear to me (Appendix H). I could not understand what probability distributions are being expressed and why. In particular, how is the term in Eq 38 approximated by the one in the first line of Eq 41. The paragraph after Eq 40 was also unclear.
- It is not discussed how this metric for evaluating interaction strength compares to something conceptually simpler like the Hessian \\(\nabla^2_{i,j} L\\) which directly measures the dependence of the network's loss on pairs of input variables, and its magnitude is proportional to the interaction strength.
- The paper mentions that an advantage of the proposed loss is that the weight \\(\lambda\\) applied to the interaction loss can be explicit controlled, whereas the strength of dropout cannot be controlled (Section 4 "advantages", "Unlike the interaction loss, people cannot explicitly control the strength of dropout .."). This does not seem correct. The dropout probability provides such as control mechanism for dropout.
- For the experimental results in Table 3, it is not mentioned what value of the dropout probability was used, whether this value was tuned for each architecture, and which network layers was dropout applied in. These factors can have a significant impact on overall performance. On the other hand, the \\(\lambda\\) parameter for the proposed interaction loss is tuned. So the resulting comparison is not fair.
- It is not clear what additional insight this metric provides about dropout, beyond confirming what is intuitively apparent : that having randomly dropped neurons will make it harder for the network to learn high-order interactions.

Other comments and suggestions:
- The introduction includes a discussion around Banzhaf value, without describing what it means. The concept of Banzhaf value might be new to many readers in the ML community. I would suggest including a short explanation to give some intuition about what it means, before discussing it in more detail.
- " the output of the DNN corresponds to the score f" : would it make sense to say that (negative) loss corresponds to the score f, rather than the output of the network ?
- "award" -> "reward" or "utility" ? (I'm not familiar with game theory literature, so I'm not sure if "award" is a commonly used term there).
- The title of the paper is a bit misleading as it seems to suggest that the paper is about using dropout in Game theory (i.e. solving problems in game theory using dropout).

Post rebuttal
The authors addressed the concerns around the clarity of the paper and added useful additional experiments. I will increase my score to 7.

---

> ### Author Response · Authors · 2020-11-20
> **Response to Reviewer #1 (Part 1)**
>
> Thank you very much for your careful review and constructive comments. We try our best to answer all your concerns.
>
> ---
>
> Q1: “The proposed metric is hard to compute exactly since it requires summing over exponentially many terms, each term requiring a forward prop through the network.”
>
> A: A good question, but the proposed metric can be efficiently estimated. Please see Appendix G for details. Although computing the exact interaction value is an NP-hard problem (this is also a typical challenge for all methods based on Shapley value (Castro et al., 2009; Lundberg et al. 2018; Ancona et al., 2019), in this paper, we used a sampling-based method to approximate the interaction, which significantly reduced the computational cost.
>
> Besides, we have conducted a **new experiment** to prove the high accuracy of such an approximation of the interaction. Experimental results showed that when the sampling number was larger than $500$, the approximated interaction was stable and accurate enough. Thus, we set the sampling number to $500$ in all experiments. Meanwhile, computing the interaction using such a sampling number was far more efficient than the original NP-hard computation of the interaction. Please see Appendix M for details.
>
> Q2: About the clarity in Appendix H: “The assumptions made in computing this metric approximately seem unclear (Appendix H)”
>
> A: Sorry for the confusion. We have further clarified the derivation by adding more details in Appendix H. Please see Appendix H for details.
>
> ---
>
> Q3: “It is not discussed how this metric for evaluating interaction strength compares to something conceptually simpler like the Hessian.”
>
> A: A good question. The Hessian matrix is another typical way to define interactions. However, the standard way of using the Hessian matrix is not to directly use the raw Hessian value $\nabla^2_{i,j}L$, but to use the Integrated-Hessian value that has been proposed in (Janizek et al., 2020). To this end, we have cited the paper in the related-work section, and added **new experiments** for more analysis.
>
> Janizek et al., (2020) proposed the Integrated-Hessian value to measure interactions, based on Integral Gradient (Sundararajan et al., 2017). Integral Gradient is used to compute an importance value for each input variable w.r.t. the DNN. Given an input vector $x\in\mathbb{R}^n$, Integrated Hessians measures the interaction between input variables (dimensions) $x_i$ and $x_j$ as the numerical impact of $x_j$ on the importance of $x_i$. To this end, Janizek et al., (2020) used Integral Gradient to compute Integrated Hessians. More specifically, given a DNN $f$ and two input variables $x_i$ and $x_j$, the interaction between $x_i$ and $x_j$ is measured as follows.
>
> $I(i,j)=(x_i-x’_i)(x_j-x’_j)\times\int _{\beta=0}^1\int _{\alpha=0}^1\alpha\beta\frac{\partial^2 f(x’+\alpha\beta(x-x’))}{\partial x_i \partial x_j}d\alpha d\beta,$
>
> where $x’$ is a baseline image.
>
> Theoretically, the Integrated-Hessian value is not equal to our interaction metric, and has different theoretic foundations with our interaction metric. Our interaction metric defines the interaction based on the Shapley value (a game-theoretic importance). In comparison, the importance value used to define the Integrated-Hessian value is computed using the Integral Gradient. Strictly speaking, the Integral-Gradient importance is highly related but not equal to the Shapley value (importance). The Shapley value is the unique importance that satisfies *the linearity property, the dummy property, the symmetry property, and the efficiency property* (Shapley, 1953; Weber, 1988; Ancona et al., 2019).
>
> 1. Compared with the raw Hessian value $\nabla^2_{i,j}L$, the Integrated-Hessian value integrates over multiple high-order gradients to generate the final Integrated-Hessian value.
>
>    To this end, we conducted an **additional experiment** to show the difference between the raw Hessian value $\nabla^2_{i,j}L$, the Integrated-Hessian value, and our interaction metric. Experimental results showed that these metrics generated different values of interactions, because these metrics defined different types of interactions from fully different perspectives. Please see Appendix L for details.
>
> 2. Unlike our interaction metric, the computation of Integrated-Hessian values requires the DNN to use the SoftPlus operation to replace the ReLU operation. The computation of our interaction metric does not have such a requirement.

---

> ### Author Response · Authors · 2020-11-20
> **Response to Reviewer #1 (Part 2)**
>
> Q4: The reviewer doubted our claim that the strength of dropout cannot be controlled.
>
> Q5: The reviewer asked about the dropout settings in Table 3, and suggest to study the performance of different dropout rates.
>
> A to Q4&Q5: Thanks. The dropout rate could control the strength of the dropout, when we consider the utility of the dropout as a penalty of interactions. However, we found that the interaction loss exhibited a stronger power to reduce the interaction.
>
> We have conducted a **new experiment** to compare the strength of penalizing interactions between the dropout with different dropout rates and our interactions loss. We trained DNNs with different dropout rates. When the dropout rate was equaled to 0.5, the DNN had the best performance. We found that when the dropout rate was 0.9, the interaction strength was penalized most, but its power of reducing the interaction was still significantly less than the interaction loss. This proved that the interaction loss was more effective than the dropout. Please see Appendix J.1 for more details about the experiment.
>
> By the way, the dropout rate in Table 3 was set as 0.5. We have clarified it in the tenth line of the third paragraph on Page 8.
>
> ---
>
> Q6: “It is not clear what additional insight this metric provides about dropout, beyond confirming what is intuitively apparent.”
>
> A: A good question. Previous studies (Hinton et al., 2012; Krizhevsky et al., 2012; Srivastava et al., 2014)  claimed that the dropout would decrease the dependence between feature variables just based on their intuitive understanding of dropout. However, these studies did not mathematically model the variable dependence or theoretically proved its relationship with the dropout.
>
> In comparison, we model the dependence using the interaction based on game theory, and build up a mathematical connection between the variable dependence and the dropout. Please see the last paragraph of the introduction section for details.
>
> Besides, we propose the interaction loss, which overcomes shortcomings of the dropout. Nevertheless, we have revised the paper to further clarify this issue. Please see the last six lines in the first paragraph on Page 8 for details.
>
> ---
>
> Q7: “The introduction includes a discussion around Banzhaf value, without describing what it means.” “I would suggest including a short explanation to give some intuition about what it means, before discussing it in more detail.
>
> A: Thanks. We have followed the suggestion to provide more discussions about the Banzhaf value in Section Introduction. Please see the first four lines on Page 2 for details.
>
> The Banzhaf value (Banzhaf III, 1964)  is another metric to measure the importance of each input variable in game theory. Unlike the Shapley value, the Banzhaf value assumes that each input variable independently participates in the game with the probability of 0.5.
>
> ---
>
> Q8: Ask about the setting of the score $f$. “Would it make sense to say that (negative) loss corresponds to the score f, rather than the output of the network?”
>
> A: A good question. We have discussed this issue in the paragraph “Understanding DNNs via game theory” on Page 3. The setting of the score $f$ is an open problem, and [cite1] has investigated this issue. If the DNN has a scalar output, then the score $f$ can be set as the scalar output. For the DNN for multi-category classification, the score $f$ can be set as the output score of the predicted category before the softmax. Alternatively, $f$ can also be set as the loss of the DNN.
>
> In addition, we have conducted a **new experiment** using the loss of the DNN as $f$. Experimental results showed that the interaction with $f=Loss$ had similar behavior with the interaction with $f=\textrm{DNN output}$. Please see Appendix J.5for details.
>
> [cite1] Covert, I.; Lundberg, S.; Lee, S.-I. 2020. Feature Removal Is a Unifying Principle for Model Explanation Method. arXiv: 2011.03623.
>
> ---
>
> Q9: About the paper writing: “‘award’ -> ‘reward’ or ‘utility’?”
>
> A: Thanks. We have followed the suggestion to polish the language.
>
> ---
>
> Q10: “The title of the paper is a bit misleading as it seems to suggest that the paper is about using dropout in Game theory.”
>
> A: Thank you. We have followed the suggestion to revise the title of the paper as “Theoretical Understanding of Dropout and Its Alternative Regularization.”

---

### Official Review · AnonReviewer3 · 2020-10-29
**AnonReviewer3**

**Rating:** 7
**Confidence:** 5

**Review:**

*Paper Summary*
The authors provide a novel interpretation of dropout regularization using Banzhaf interactions, a tool from game theory.

*Pros*
* The authors are able to mathematically prove that dropout is capable of suppressing neural co-adaptations, the latter being one of the reasons for overfitting. Visualizations are also provided in this respect on a dataset for face analysis.
* Through their mathematical analysis, authors are able to improve upon the classical dropout training, by making it more compatible with batch normalization, so that these two classical regularization strategies show a better complementarity.

*Cons*
* Some of the results does not read well, like Table 3 or Figure 4, but this is really minor and fixable

*Preliminary Evaluation*
I believe that the overall analysis provided by authors is complete and interesting, so I am akin to call for a full acceptance of the paper which I deem suitable for such a venue like ICLR. In order to improve their paper, I would encourage authors to better investigate over the following aspect: since many times authors established a principled connections between dropout and neural activations, it would be very interesting to discuss the relationship with the present work and another paper [Gomez et al, Targeted Dropout, NeurIPS Workshops 2018] in which a computational variant of dropout is proposed, such that the dropout rate depends upon neural activations.

*Post-Rebuttal Evaluation*
I have carefully read the response provided by authors and checked the revised manuscript. I confirm my preliminary acceptance rate.

---

> ### Author Response · Authors · 2020-11-20
> **Response to Reviewer #3**
>
> Thank you very much for your careful review and constructive comments. We try our best to answer all your concerns.
>
> ---
>
> Q1: **About the clarity of Table 3 and Figure 4.** “Some of the results does not read well, like Table 3 or Figure 4, but this is really minor and fixable.”
>
> A: Thank you for your comments. We added more discussions on results in Table 3 and Figure 4, as follows, which provided some new insights. Please see the last two paragraphs on Page 8 for details.
>
> Table 3 shows that the accuracy of DNNs trained with different values of $\lambda$ and the accuracy of DNNs trained with dropout. The accuracy of DNNs usually increased along with the $\lambda$ at first, and then decreased when the value of $\lambda$ continued to increase. This could be explained as follows. The interaction loss alleviated the significance of over-fitting when the weight $\lambda$ was small, and then caused under-fitting when the weight $\lambda$ was large. Furthermore, the DNN trained using the interaction loss usually outperformed the DNN trained using the dropout, when the weight $\lambda$ was properly selected. This verified that the interaction loss would improve the utility of the dropout.
>
> Figure 4 shows curves of the interaction and the accuracy of DNNs during the learning process with different weights $\lambda$ of the interaction loss. First, we found that the interaction decreased along with the increase of the weight $\lambda$, which verified the effectiveness of the interaction loss. Second, when the weight $\lambda$ was properly selected, the DNN trained with the interaction loss usually outperformed the DNN trained with dropout. Besides, the interaction of the DNN trained using the interaction loss was much lower than the interaction of the DNN trained using the dropout. I.e., the interaction loss was more effective to suppress the interaction and boost the performance than the dropout.
>
> ---
>
> Q2: Suggest to discuss another paper related paper [Gomez et al., Targeted Dropout, NeurIPS Workshops 2018].
>
> A: Thank you. We have cited this paper (Gomez et al. 2018), and discussed it in the related-work section. (Gomez et al. 2018)proposed the targeted dropout, which only randomly dropped variables with low activation values, and kept variables with high activation values. In comparison, our interaction metric quantified the interaction between all pairs of variables considering all potential contexts, which were randomly sampled from all variables. Please see the last fifth line in the third paragraph on Page 2 for the discussion.
>
> Aidan N. Gomez, Ivan Zhang, Kevin Swersky, Yarin Gal, Geoffrey E. Hinton. Targeted Dropout. NeurIPS Workshop 2018.

---

### Official Review · AnonReviewer2 · 2020-10-30
**An interesting explanation of dropout from the lens of game theoretic interactions, experiments require some important improvements.**

**Rating:** 7
**Confidence:** 4

**Review:**

Summary.

This paper aims to explain dropout from the lens of game theoretic interactions. Let x denote the input of a deep neural net (DNN), intuitively, the interaction between two variables x_i and x_j quantifies how much the presence/absence of the j-th variable affects the contribution of the i-th variable to the output of the DNN. With the above definition in place, the authors show theoretically and empirically that dropout reduces the interactions between input variables of DNNs. As this type of interactions turn out to be strongly correlated with overfitting, the authors suggest that dropout alleviates overfitting by reducing interactions between input variables (or activation units) of DNNs. Based on this understanding of dropout, an alternative regularization technique is proposed, which explicitly penalizes pairwise interactions between variables.

Strengths.

1. The paper is well written and clearly presented.

2. Although it is already well known (or at least widely accepted) in the community that dropout reduces dependencies among activation units in DNNs, the explanation of dropout from the perspective of game theoretic interactions is interesting, and it is supported both theoretically as well as empirically.

Weakness.

1. Hyperparameter settings (e.g., optimization-related ones) to reproduce the results are not provided. It is not clear what dropout rate was used in the experiments. Is it 0.5? In all cases, different dropout rates should be investigated before claiming the superiority of the proposed interaction loss (regularization) over dropout.

2. Experiments are curried out on only one task (classification), one type of data (images), and one family of DNNs (convolutional neural nets). However, the paper draws quite general conclusions regarding the understanding of dropout from the perspective of game theoretic interactions. Therefore, considering at least one more task involving a different type of data and another family of DNNs would reinforce the findings of this paper.

3. Computational time analysis of the proposed interaction loss and training time comparisons with dropout are lacking.

Additional comments

1. Dropout is used both at convolutional and at fully connected layers. However, one can argue that applying dropout to convolutional layers does not make sense owing to the sparsity of connections in this type of layers.

2. I would recommend revising the title of the paper. What is proposed is more of an alternative regularization form to dropout than an improvement for the latter.

---

> ### Author Response · Authors · 2020-11-20
> **Response to Reviewer #2**
>
> Thank you very much for your careful review and constructive comments. We try our best to answer all your concerns.
>
> ---
>
> Q1: “Hyperparameter settings (e.g., optimization-related ones) to reproduce the results are not provided. It is not clear what dropout rate was used in the experiments.”
>
> A: Thank you. First, in experiments, we set the dropout rate as 0.5. Please see the tenth line of the third paragraph on Page 8 for the setting of the dropout rate.
>
> Second, we have followed your suggestions to conduct a **new experiment** to further study the performance of different dropout rates. We found that when the dropout rate was 0.5, the DNN achieved the best performance. The dropout rate of 0.9 usually penalized more interactions than other dropout rates. Nevertheless, DNNs trained using the interaction loss still penalized more interactions than DNNs trained with the dropout in most cases. Please see Appendix J.1 for more details about the experiment.
>
> ---
>
> Q2: Suggestions for experiments on more datasets and more DNNs. “Considering at least one more task involving a different type of data and another family of DNNs would reinforce the findings of this paper.”
>
> A: Thank you. We have followed your suggestions to conduct **new experiments**, in which we learned the BERT on the SST-2 dataset (linguistic data) for the binary sentiment classification task. We further compared our method with the traditional dropout based on the BERT model. Additional experimental results showed that the interaction loss successfully decreased the interaction modeled by DNNs, and boosted the performance. Please see Appendix J.2 for details about the additional experiment.
>
> Besides the above new DNN, the proposed interaction loss has been compared with the traditional dropout on the AlexNet, VGG-11, VGG-13, VGG-16, VGG-19, ResNet-18, and ResNet-34 trained on the CIFAR-10 dataset for image classification, the Tiny ImageNet dataset for image classification, and the CelebA dataset for the gender estimation. All these experiments have proved that the interaction loss is an effective method to suppress the interaction modeled by DNNs, and boost the performance of DNNs.
>
> ---
>
> Q3: “Computational time analysis of the proposed interaction loss and training time comparisons with dropout are lacking.”
>
> A: Thanks. We have followed the suggestion to conduct an **additional experiment**, in which we separately recorded the time cost of training a DNN with the proposed interaction loss and the time cost of training a DNN with the dropout. Our interaction loss increased the time cost, but it was still of specific value considering its good performance and the harmony with the batch-normalization operation. Please see Appendix J.4 for the computational cost.
>
> ---
>
> Q4: Concerns about applying dropout on convolutional layers. “Applying dropout to convolutional layers does not make sense owing to the sparsity of connections in this type of layers.”
>
> A: A good question, but there have been various studies, which investigated behaviors of dropout on the convolutional layers or on the input (Konda et al. 2016;cite1;cite2). These studies have shown that in spite of the feature sparsity, theoretically, the dropout is still applicable to convolutional layers.
>
> Nevertheless, we have conducted **additional experiments** to apply the dropout on more fully-connected layers. We found that DNNs trained with dropout on the convolutional layer had similar performance with DNNs trained with dropout on fully-connected layers. Thus, dropout was applicable to convolutional layers. Please see Appendix J.3 for details of additional experiments.
>
> [cite 1] J. Tompson, R. Goroshin, A. Jain, Y. LeCun, and C. Bregler, “Efficient object localization using convolutional networks.” in IEEE CVPR. IEEE, 2015, pp. 648–656.
>
> [cite 2] S. Park and N. Kwak, “Analysis on the dropout effect in convolutional neural networks,” in Asian Conference on Computer Vision. Springer, 2016, pp. 189–204.
>
> ---
>
> Q5: “I would recommend revising the title of the paper. What is proposed is more of an alternative regularization form to dropout than an improvement for the latter.”
>
> A: Thank you. We have followed your suggestion to revise the title as “Theoretical Understanding of Dropout and Its Alternative Regularization.”

---

> > ### Comment · AnonReviewer2 · 2020-11-24
> > **Response to the authors**
> >
> > Thank you for the response. The authors have addressed most of my original concerns and revised the paper accordingly. I therefore increase my score to reflect these updates. That said, there are still two remaining concerns.
> >
> > 1. Some important experimental settings such as optimization-related hyperparameters are lacking.
> >
> > 2. Technically dropout is applicable to convolutional layers. However, as the latter have substantially less parameters, compared to fully connected layers, they are less prone to overfitting, and they would require less regularization.  Moreover, as the feature maps are expected to encode spatial relationships, the activations can naturally exhibit high correlation, which would make dropout not suitable in this case. In particular, I would recommend revising section J.3 in the appendix to address the following concerns.
> >
> >     a. It is obvious that dropout is technically applicable to convolutional layers. Maybe what you are trying to investigate in this experiment is the utility or the benefit of applying dropout to convolutional layers.
> >
> >     b. The results of this section are not convincing and not enough to claim that it is beneficial to apply dropout to convolutional layers. In one case (AlexNet), applying dropout before the 1st or 2nd fully connected layer seems better than applying it before the conv layer. In the other case (VGG-11) we observe the opposite behavior, however, in this case not using dropout seems even better (as reported in Table 4), which suggests an underfitting situation.

---

> > > ### Author Response · Authors · 2020-11-25
> > > **Responses to additional questions from Reviewer #2**
> > >
> > > Thank you very much for your changing the rating to 6. Meanwhile, we are pleased to answer your new concerns.
> > >
> > > ---
> > >
> > > Q1: "Some important experimental settings such as optimization-related hyper-parameters are lacking."
> > >
> > > A: Thank you. We have clarified experimental settings of optimization-related hyper-parameters in the last paragraph of Section 4 and the paragraph under Equation (36), which include $\alpha$ in Equation (9), $s$ in Equation (3), and the sampling time of $s$ in Equation (36).
> > >
> > > We have also introduced experimental settings of more detailed hyper-parameters in the last paragraph of Section 4, which include the choice of optimizers, the learning rate, the momentum, etc. All baseline models (including DNNs trained with the interaction loss and DNNs trained using dropout) were trained using the same hyper-parameters on each dataset, which enabled fair comparisons.
> > >
> > > Furthermore, we have also uploaded the **source code** as the supplementary material to ensure the reproducibility.
> > >
> > > ---
> > >
> > > Q2: About the experiment in Appendix J.3. "Maybe what you are trying to investigate in this experiment is the utility or the benefit of applying dropout to convolutional layers." "The results of this section are not convincing and not enough to claim that it is beneficial to apply dropout to convolutional layers."
> > >
> > > A: Thanks, but this concern may be raised by the misunderstanding of the conclusion in the paper. In the second paragraph of Page 9, we claimed that there was not a clear principle about the suitable position for dropout. Besides, in Appendix J.3, we claimed that the dropout operation was applicable to the convolutional layer without obvious disadvantages w.r.t. applying the dropout to FC layers, but we had never claimed that it was "beneficial to apply dropout to convolutional layers."
> > >
> > > Besides, experimental results in Table 2 and Table 6 also supported the above two conclusions, respectively.
> > >
> > > Nevertheless, we have revised the paper writing in Appendix J.3 to avoid the confusion about our claim. Thank you very much for your careful review.

---

### Author Response · Authors · 2020-11-20
**Summary of changes**

We would like to thank all the reviewers for their careful reviews and valuable comments. Based on reviewers’ feedback, we have updated the paper, with the following revisions:

1. We have conducted a **new experiment** to further study the performance of different dropout rates. DNNs trained using interaction loss still decrease more interactions than DNNs trained with the dropout in most cases. Please see Appendix J.1 for details about the experiment.
2. We have conducted **new experiments**, in which we learned the BERT on the SST-2 dataset (linguistic data) for the binary sentiment classification task. Experimental results showed that the interaction loss successfully decreased the interaction modeled by DNNs, and boosted the performance. Please see Appendix J.2 for details about the additional experiment.
3. We have conducted an **additional experiment**, in which we recorded the time cost of training a DNN with the proposed interaction loss and the time cost of training a DNN with the dropout. Our interactions loss increased the time cost, but it was still of specific value considering its good performance and the harmony with the batch-normalization operation. Please see Appendix J.4 for the computational cost.
4. We have conducted **additional experiments** to apply the dropout on more fully-connected layers. We found that DNNs trained with dropout on the convolutional layer had similar performance with DNNs trained with dropout on fully-connected layers. Thus, the dropout was applicable to convolutional layers. Please see Appendix J.3 for details of additional experiments.
5. We have conducted a **new experiment** using the loss of the DNN as $f$. Experimental results showed that the interaction with $f=Loss$ had similar behavior with the interaction with $f=\textrm{DNN output}$. Please see Appendix J.5 for details.
6. We have conducted an **additional experiment** to show the difference between the raw Hessian value $\nabla^2_{i,j}L$, the Integrated-Hessian value, and our interaction metric. Experimental results showed that these metrics generated different values of interactions, because these metrics defined different types of interactions from fully different perspectives. Please see Appendix L for more details.
7. We have conducted a **new experiment** to prove the high accuracy of the sampling-based approximation of the interaction. Experimental results showed that when the sampling number was larger than $500$, the approximated interaction was stable and accurate enough. Thus, we set the sampling number to $500$, which significantly reduced the time cost in all experiments in the paper. Please see Appendix M for details.
8. We have revised the title as “Theoretical Understanding of Dropout and Its Alternative Regularization.”
9. We have added more discussions on results in Table 3 and Figure 4, as follows, which provide some new insights. Please see the last two paragraphs on Page 8 for details.
10. We have cited another related paper (Gomez et al. 2018), and discussed it in the related-work section. Please see the the last fifth line in the third paragraph on Page 2 for the discussion.
11. We have further clarified the derivation by adding more details in Appendix H. Please see Appendix H for details.
12. We have added more discussion about the Hessian matrix. Please see Appendix L for details.
13. We have clarified the dropout rate used in experiments in the tenth line of the third paragraph on Page 8
14. We have further clarified the contribution of this paper. Please see the last paragraph of the introduction section for details.
15. We have provided more discussions about the Banzhaf value in Section Introduction. Please see the first four lines on Page 2 for details.
16. We have provided more discussions about the setting of the score $f$. Please see the paragraph “Understanding DNNs via game theory” on Page 3 for details.
17. We have polished the language.

---

### Decision · Program_Chairs · 2021-01-07
**Final Decision**

**Decision:**

Accept (Poster)

**Comment:**

The paper introduces a game-theoretic framework to improve our understanding of dropout. All reviewers appreciated the contribution of the paper. While they had a number of questions/suggestions, almost all of them were adequately addressed. Three reviewers are satisfied and recommend acceptance, while a lone reviewer is on the fence, he/she admits he/she is less knowledgeable about game theory. Overall, I think this paper makes a solid contribution to ICLR.